# Dietary vitamin E reaches the mitochondria in the flight muscle of zebra finches but only if they exercise

Clara Cooper-Mullin[1]*, Wales A. Carter[1], Ronald S. Amato[2], David Podlesak[2], Scott R. McWilliams[1]

**1** Department of Natural Resources Science, University of Rhode Island, Kingston, Rhode Island, United States of America, **2** Los Alamos National Laboratory, Los Alamos, New Mexico, United States of America

☯ These authors contributed equally to this work.

* ccooper-mullin@uri.edu

**Data Availability Statement:** The data underlying the results presented in the study are available from Dryad: https://datadryad.org/stash/dataset/doi:10.5061/dryad.7wm37pvsv.

## Abstract

Whether dietary antioxidants are effective for alleviating oxidative costs associated with energy-demanding life events first requires they are successfully absorbed in the digestive tract and transported to sites associated with reactive species production (e.g. the mitochondria). Flying birds are under high energy and oxidative demands, and although birds commonly ingest dietary antioxidants in the wild, the bioavailability of these consumed antioxidants is poorly understood. We show for the first time that an ingested lipophilic antioxidant, α-tocopherol, reached the mitochondria in the flight muscles of a songbird but only if they regularly exercise (60 min of perch-to-perch flights two times in a day or 8.5 km day$^{-1}$). Deuterated α-tocopherol was found in the blood of exercise-trained zebra finches within 6.5 hrs and in isolated mitochondria from pectoral muscle within 22.5 hrs, but never reached the mitochondria in caged sedentary control birds. This rapid pace (within a day) and extent of metabolic routing of a dietary antioxidant to muscle mitochondria means that daily consumption of such dietary sources can help to pay the inevitable oxidative costs of flight muscle metabolism, but only when combined with regular exercise.

## Introduction

Mitochondria are the key sites of energy metabolism in all oxygen consuming organisms. However, an inevitable byproduct of this metabolism are reactive species (RS) produced in the mitochondria [1]. When produced at low levels, RS act as important signaling molecules in a variety of physiological processes, but also interact with mitochondria, cells and tissues to cause oxidative damage [1, 2]. Although animals have a complex endogenous antioxidant system that can and does respond to oxidative challenges [3], upregulating this endogenous system likely costs energy and time [4]. Consumption of dietary antioxidants (e.g. tocopherols, polyphenols) can mitigate this cost [4] and can play an essential role in preventing oxidative damage—especially during energy-demanding life events such as growth, exercise (e.g.

**Funding:** SRM: This work was supported by the National Science Foundation [IOS-0748349] https://www.nsf.gov/ and the United States Department of Agriculture [RIAES-538748] https://www.usda.gov/. The funders had no role in study design, data collection and analysis, decision to publish, or preparation of the manuscript.

**Competing interests:** The authors have declared that no competing interests exist.

migration), breeding, or when faced with an immune challenge [5–7] Supplementing the diets of humans (e.g., athletes), birds, or mammals with certain antioxidants prior to high-intensity exercise-training protects against oxidative damage to lipids [8, 9] and muscles [10]. A wide variety of animals prefer foods with certain antioxidants [11–13] and these micronutrients are abundant in many foods consumed by wild animals [11, 12, 14, 15]. Although the functional importance of dietary antioxidants to animal health and performance is recognized [16–18], no previous studies have demonstrated that consumed antioxidants are transported to a major site of RS production, the mitochondria (but see: [19], and whether this depends at all on the physiological state of the animal.

Flying birds offer a particularly relevant system for investigating how exercise affects the bioavailability and efficacy of dietary antioxidants, given the high energetic costs of flight [20–22], birds' reliance on fats as fuel [23], and their preference for foods rich in antioxidants [11, 13, 24–27]. Flight is a particularly energetically expensive form of exercise and is primarily fueled by catabolizing stored fat—nature's most energy dense, yet oxidatively vulnerable metabolic fuel [28, 29]. Fats, especially polyunsaturated fats, are particularly vulnerable to oxidative damage due to their low oxidative potential and once damaged form lipid radicals that can then attack other lipids, creating a cascade of damage [29]. Therefore, storing and catabolizing fats for flight puts birds at a higher risk of oxidative damage and an increased need for antioxidants (endogenous and/or dietary). Additionally, due to the nature of aerodynamics, perch-to-perch flights like the flight performed while foraging increase an individual's metabolic rate up to 30 times basal metabolic rate (BMR) [22, 30] and, therefore, likely increase RS production and impose a high oxidative load [29]. Likewise, long-duration, sustained flights by migrating birds, some of the world's most incredible athletes, require prolonged periods operating at high metabolic rates (>10X BMR) while not eating or drinking [31–33]. Birds regularly consume lipophilic and hydrophilic antioxidants from fruits during fall migration [11, 13, 24, 34], but evidence that these antioxidants protect against damage by RS during high-intensity exercise is murky due in part to the complexity of the antioxidant system [3, 29, 35], as well as our inability to directly measure rates of RS production *in vivo* that would result in damage [36, 37]. Further, however common in their diets, the bioavailability of dietary antioxidants in songbirds is largely unknown (although see: [38]). Importantly, if and only if dietary antioxidants are assimilated by the gut and then transported to the mitochondria, can they effectively alleviate the oxidative costs associated with energy-demanding life events such as flying.

One of the eight stereoisomers of vitamin E (α-Tocopherol) is an essential dietary antioxidant (i.e., cannot be produced *de novo* by any vertebrate and so must be consumed) that acts as a chain-breaker in the propagation of lipid peroxidation [1, 29, 39], and once oxidized can be recycled by vitamin C *in vivo* [1]. The livestock and poultry industries have long acknowledged vitamin E to be crucial for preventing oxidation of meat, and have focused on understanding how to use dietary supplementation to increase subcellular concentrations of vitamin E [40, 41]. Holstein and beef steers that consumed vitamin E-supplemented diets had higher tocopherol concentrations in muscle mitochondrial fractions as compared to steers not given dietary vitamin E [42] although this has not been shown in wild mammals or birds. Additionally, dietary supplementation increased α-tocopherol content of the subcellular membranes of various porcine and chicken muscles [43, 44]. Although mitochondria are known to have high α-tocopherol concentrations, especially in the inner mitochondrial membrane [45, 46], the timing and deposition of α-tocopherol from the diet to plasma to mitochondria in different tissues remains largely unknown, especially in passerine birds. Further, as α-tocopherol can be recycled [1], it is unclear whether an observed increase in α-tocopherol concentrations in subcellular fractions are a direct result of deposition from the diet or an indirect effect on the recycling of α-tocopherol by vitamin C.

To our knowledge, no previous studies have directly demonstrated (a) that α-tocopherol from the diet is metabolically routed [47, 48] to the muscle mitochondria in a volant species with relatively higher metabolic rates and potential RS production than mammals or fish, and (b) how the metabolic routing is directly affected by very high-energy demanding exercise, in this case in actively-flying birds [31, 49]. This is the first study in passerines to trace α-tocopherol from the diet into circulation and then mitochondria of the primary flight muscle, and the first in birds to examine whether exercise changes the absorption and deposition of α-tocopherol into tissues and organelles. Given the high energy and oxidative costs of flying in birds, elucidating the pace and extent of metabolic routing of a commonly consumed dietary antioxidant would reveal the potential of such dietary sources for paying these oxidative costs. Specifically, we tested the following hypothesis: The rate at which dietary antioxidants are routed to tissues depends on exercise with faster routing in flown vs. sedentary birds. We also provide the first estimates of the time course of metabolism of lipid-soluble dietary antioxidants in sedentary and flying birds.

## Methods

### Exercise training

We performed this experiment on zebra finches (*Taeniopygia guttata*) obtained from a known-age captive population of birds at Sacred Heart University [50, 51]. This captive population allowed us more control over potential variation in age [52, 53], sex [54, 55], gastrointestinal microbiome composition [56], and diet [1, 36, 57], which can influence absorption of dietary antioxidants or the response of the antioxidant system to exercise. Zebra finches are also social and easily trained to fly in large numbers in a flight arena, and their use in this study allowed us to avoid sacrificing wild migratory songbirds [58].

The birds used in this experiment were part of a larger experiment focused on lipid turnover rates and changes in the antioxidant system in exercised and non-exercised birds [50, 51]. The finches (n = 65, all hatch-year) were randomly assigned to an exercise-trained treatment group (N = 33, 16 males, 17 females) or an unexercised group (N = 32, 15 males, 17 females). Prior to exercise-training, the 65 zebra finches were kept in same-sex aviaries (2.1 x 0.9 x 1.8 m = L x W x H) for 8 weeks on a 14 h:10 h light:dark cycle under full-spectrum light, with lights on at 06:00. The aviaries were made of wire mesh and birds were provided perches for natural movement. Birds were banded with an aluminum numbered band on their right leg for individual recognition with one color band on the left leg to indicate treatment (exercised males = red; unexercised males = green). During those 8 weeks, birds acclimated to the aviaries, and to a standard mixed-seed diet (Abbaseed #3700, Hillside, NJ USA) composed mainly of canarygrass (*Phalaris canariensis*) and had *ad libitum* access to water, grit, and cuttlebone. All zebra finches were given fresh kale, a source of vitamin C and α-tocopherol, once weekly [50, 51, 59].

The exercise-trained zebra finches were subjected daily to two one-hour periods of stop-and-go perch-to-perch flights (11:00–12:00 and 13:30–14:30) in a 6 (*l*) x 3 (*w*) x 2 m tall flight arena in an adjacent room from the aviaries for up to 70 days (see [36, 50, 51, 60, 61]). Each day during the 48 or 70 days of exercise-training, zebra finches flew between perches located 4 m apart in opposite corners of the arena. A cloth wall partially divided the arena in the middle such that birds departing from a given perch flew to one side of the arena and returned on the opposite side. During the two 1 h sessions each day, a handler walked clockwise around the arena for 300 laps per hour, which resulted over the two hrs in about 1200 8-m flights or approximately 8.5 km per day. This type of short-burst flight incurs energetic costs that are approximately 3x higher than sustained flight [22] and respiratory quotients for zebra finches

exposed to perch-to-perch flights were indicative of primarily burning fats (0.75 ± 0.01) [49], a fuel type that increases the potential for oxidative damage [29]. During the two 1 h exercise periods, we removed food and water from the cages housing unexercised birds to ensure both treatment (exercise-trained) and control (unexercised) birds were similarly fasted. Although unexercised birds were able to move around unhindered within their cages, they were never able to fly more than 2.1 m at a time and occasional observations of birds in the cages suggest that they spent most of their time perched and not flying.

## Blood sampling, mitochondrial isolation and mass spectrometry

We randomly drew 19 males from the larger flight-exercised and unexercised groups of zebra finches (n = 65) for this experiment. We sampled only males because a previous study found differences in oxidative damage associated with flight-training between males and females [29]. Due to time constraints associated with the need to keep the tissues cold and alive during the process of isolating mitochondria from pectoral muscle (see below), we sacrificed subsets of males in each treatment group at two different time points as follows. After 48 days of daily exercise-training, we randomly selected 8 male zebra finches from the unexercised group and 4 male zebra finches from the exercised group to be gavaged with 150 μL (or 0.200 μmoles) of deuterated α–tocopherol (d$_6$-RRR- α–tocopherol acetate, a gift from Dr. John Lodge, St. Thomas' Hospital, London, hereafter: d$_6$α-tocopherol or $\delta$D α-tocopherol to indicate the isotopic ratio) dissolved in olive oil. The gavaged amount of d$_6$α-tocopherol (30 IU/dose) was chosen based on recommended supplements for poultry and to avoid any negative effects of over supplementation [62, 63]. After 70 days of exercise-training, we randomly selected a second group of 4 male zebra finches from the exercised group to be gavaged with 150 μL of d$_6$α-tocopherol, and 3 male zebra finches from the unexercised group to be gavaged with 150 μL of olive oil. This latter group (olive oil only) was used to determine the natural background level of deuterated hydrogen in the blood and isolated mitochondria of zebra finches and is generally assumed to be highly stable [64, 65]. To determine a timecourse for incorporation of d$_6$α-tocopherol into blood and mitochondria, we randomly selected whether birds were gavaged the night before (22.5 hours prior to sampling) or in the morning (6.5 hours prior to sampling). We chose these time points given that previous studies on the appearance and half-life of lipid-soluble dietary antioxidants in plasma of livestock and humans demonstrated that these antioxidants appear *in vivo* within 2–30 hours of ingestion [41, 66, 67]. Directly after the second one-hr flight, we sacrificed the exercised and unexercised birds by cervical dislocation, took a blood sample from the severed carotid artery, and removed the right pectoral muscle. Blood samples were flash frozen in liquid nitrogen and stored at -80˚C until further analysis. Pectoral muscle was placed in an ice-cold high sucrose isolation buffer solution modified from Scott et al. [68]. Mitochondria from the pectoral muscle of each bird were isolated using differential centrifugation and a Percoll gradient [68–71]. Both mitochondria and blood samples were freeze dried and analyzed with a mass spectrometer to detect amount of d$_6$α-tocopherol. All care and experimental procedures were reviewed and approved by the University of Rhode Island's Institutional Animal Care and Use Committee under protocol AN11-12-009. We detected no difference in the amount of deuterated α–tocopherol in samples from males sacrificed after 48 or 70 days of flight-training so the duration of flight-training was not included in the final statistical models.

**Sample preparation.** Blood samples (n = 19) and isolated mitochondria (n = 17) were freeze dried at the University of Rhode Island and mailed to Los Alamos National Laboratories for further analysis. Upon arrival in New Mexico, blood and mitochondria samples were allowed to equilibrate with local air [72] and freeze dried again for 24 hours on non-

consecutive days (48 hours total). We weighed 150 μg of samples in duplicate into silver foil capsules before a second round of drying in a vacuum desiccator for approximately one week. We used 150 μg of blood and mitochondria based on an earlier analysis that showed sufficient δD α-tocopherol appeared in muscle tissue with this amount of sample.

**Isotope Ratio Mass Spectrometry.** We determined the isotopic composition of the samples with an Isotope Ratio Mass Spectrometry (IRMS) system, consisting of a Delta Plus IRMS (Thermo Scientific, Bremen, GE), high temperature thermal conversion (Thermo Scientific TC/EA) and Conflo IV Interface (Thermo Scientific). Samples in silver foil capsules were dropped via an autosampler into a heated glassy carbon column at 1400˚C to convert organic and inorganic constituents into $H_2$¬, $N_2$, and CO gases. All gases were swept into an isothermally packed gas chromatography (GC) column with an ultra-pure helium carrier gas and separated. The carrier plus separated sample gases were passed to the Conflo IV interface which controls the amount of sample and reference gas sent to the spectrometer. We analyzed a reference gas (ultra-pure hydrogen) twice prior to each sample analysis and compared the sample isotopic ratios to the reference gas for ratio determination expressed as a delta value by Isodat 3.0 (Thermo) software. The IRMS measured ions at mass-to-charge (m/z) ratios of 2 ($^1H_2$) and 3 ($^1H_2H$). Additionally, we analyzed several reference materials of certified isotopic composition at the beginning, end and periodically throughout the analysis. Well characterized working standards (USGS-42 Tibetan Human Hair, and USGS-43 Indian Human Hair) were simultaneously analyzed and scaled to accepted international standards (VSMOW, Vienna Standard Mean Ocean Water). Post analysis delta values were normalized to these standards. Stable isotope ratios are reported in delta-notation as parts per thousand (‰) deviations from the international standard for deuterium (δD). Therefore, sample delta values are a delta scale calibrated to the VSMOW standard of $^2H/^1H = 0.00 ± 0.3$. All samples were run in triplicate and coefficients of variation for each sample were less than 10%. Final sample sizes varied after analysis as some samples were not able to be dried completely (Blood Samples: n = 3 Background, 7 Exercised, 8 Unexercised; Mitochondria Samples: n = 3 Background, 7 Exercised, 7 Unexercised). Background mitochondria deuterium values (range: -53.29 to -45.01 ‰) and background blood deuterium values (range: -134.03 to -130.17 ‰) were stable with little variation. All care and experimental procedures were reviewed and approved by the University of Rhode Island's Institutional Animal Care and Use Committee under protocol AN11-12-009.

## Statistics

We used the *bbmle* package in R (v1.0.23.1, [73]) to conduct all analyses. We used separate linear models for whole blood and mitochondria to determine if the amount of δD-α-tocopherol in a tissue varied with treatment group (Exercised, Unexercised) as referenced by background deuterium levels for each tissue (categorical variable), the time elapsed after a bird was gavaged (continuous variable), and the amount of administered δD α-tocopherol (mL olive oil mixture; continuous variable). We used the information-theoretic approach based on Akaike's Information Criterion for small sample sizes (AICc) to select the best model for each tissue (blood, muscle). The highest ranked models were those with the highest Akaike weight ($w_i$) and a change in AICc (ΔAIC$_c$) [74]. Akaike weights represent the weight of evidence in favor of a given model being the best model and given that one of the set of candidate models must be the "best" model based on the distance (Kullback–Leibler information) between the selected model and the next candidate model [74]. We obtained coefficients (ß), t-values, and p-values for each individual coefficient for each top model with the function summary in the base R *stats* package ([75], (Table 1). We also assessed whether non-linear effects would improve the models, but found that they did not provide a better model fit. We checked whether variables

**Table 1. Model selection to explain δD-α-tocopherol values in whole blood and isolated mitochondria.**

| Blood Samples | | | | |
|---|---|---|---|---|
| Model | df | Log(L) | $\Delta AIC_c$ | $w_i$ |
| **Exercise Treatment + Time Since Gavage** | **5** | **-78.54** | **0.00** | **0.53** |
| Exercise Treatment | 4 | -81.17 | 1.33 | 0.27 |
| Null | 1 | -94.39 | 21.48 | 0.000 |
| Mitochondria Samples | | | | |
| Model | df | Log(L) | $\Delta AIC_c$ | $w_i$ |
| **Exercise Treatment * Time Since Gavage + Amount Gavaged** | **5** | **-45.31** | **0.00** | **0.94** |
| Exercise Treatment + Time since gavage | 4 | -51.20 | 6.80 | 0.03 |
| Null | 2 | -54.00 | 8.28 | 0.01 |

Degrees of freedom (df), maximized log-likelihood [log(L)], change in second-order Akaike Information Criterion for small sample sizes (AICc), and Akaike weights ($w_i$) for each of the candidate models used to determine if the amount of δD-α-tocopherol in a tissue varied with treatment group (Exercised, Unexercised) as referenced by background deuterium levels, the time that a bird was gavaged (22.5 hrs or 6.5 hrs prior to sampling), and the amount of administered δD α-tocopherol (mL olive oil mixture). Separate models were run for each tissue measured. Models with the highest rank are in bold.

met the assumptions of homogeneity of variance with Levene's tests using the *car* package (v_3.0–9, [76]), visually inspected residual plots, and did not find any noticeable deviations from homoscedasticity or normality. Results are presented as mean ± standard error (SE) and the full R code is available in supplementary materials.

We estimated mean retention time of δD-α-tocopherol in whole blood (τ) using an exponential decay function:

$$y_t = y_\infty + ae^{\left(-\frac{t}{\tau}\right)} \tag{1}$$

where $y_t$ is the measured δD α-tocopherol value of blood at time t, $y_\infty$ is the estimated asymptotic, or background, δD α-tocopherol value of the blood, *a* is the estimated difference between the δD α-tocopherol values of blood sampled 6.5 hrs after gavage and at equilibrium with background δD samples, *t* is the measured time since gavage in hours, and τ is the estimated mean retention time of δD-α-tocopherol. Since the time at which δD α-tocopherol in blood would return to background was unknown, we fit the model using a range of specified time values for the asymptote (30 to 250 hours) until the model converged, and then estimated τ from the model with the best fit. All statistics were performed with R Core Team version 4.0.2 (2020).

## Results

Whole blood δD α-tocopherol was higher than background for exercise-trained zebra finches at both 6.5 hrs and 22.5 hrs following gavage, but no time points were significantly different from background for unexercised zebra finches (Fig 1, adjusted $R^2 = 0.79$, Exercised: ß = 93.45, $t_{3,14} = 5.03$, $P < 0.001$; Unexercised: ß = 8.59, $t_{3,14} = 0.46$, $P = 0.65$; Hours Since Gavage: ß = -1.55, $t_{3,14} = -2.18$, $P = 0.05$). Deuterated α-tocopherol appeared in mitochondria isolated from pectoral muscle of exercise-trained birds within 22.5 hrs, but δD α-tocopherol was never higher than baseline in mitochondria isolated from unexercised birds (Fig 2, adjusted $R^2 = 0.70$, Interaction: ß = 1.80, $t_{3,10} = 3.65$, $P = 0.004$; Exercised: ß = -12.73, $t_{3,10} = -1.60$, $P = 0.15$; Hours Since Gavage: ß = -0.28, $t_{3,10} = -0.80$, $P = 0.44$). Estimated retention time (τ) in whole

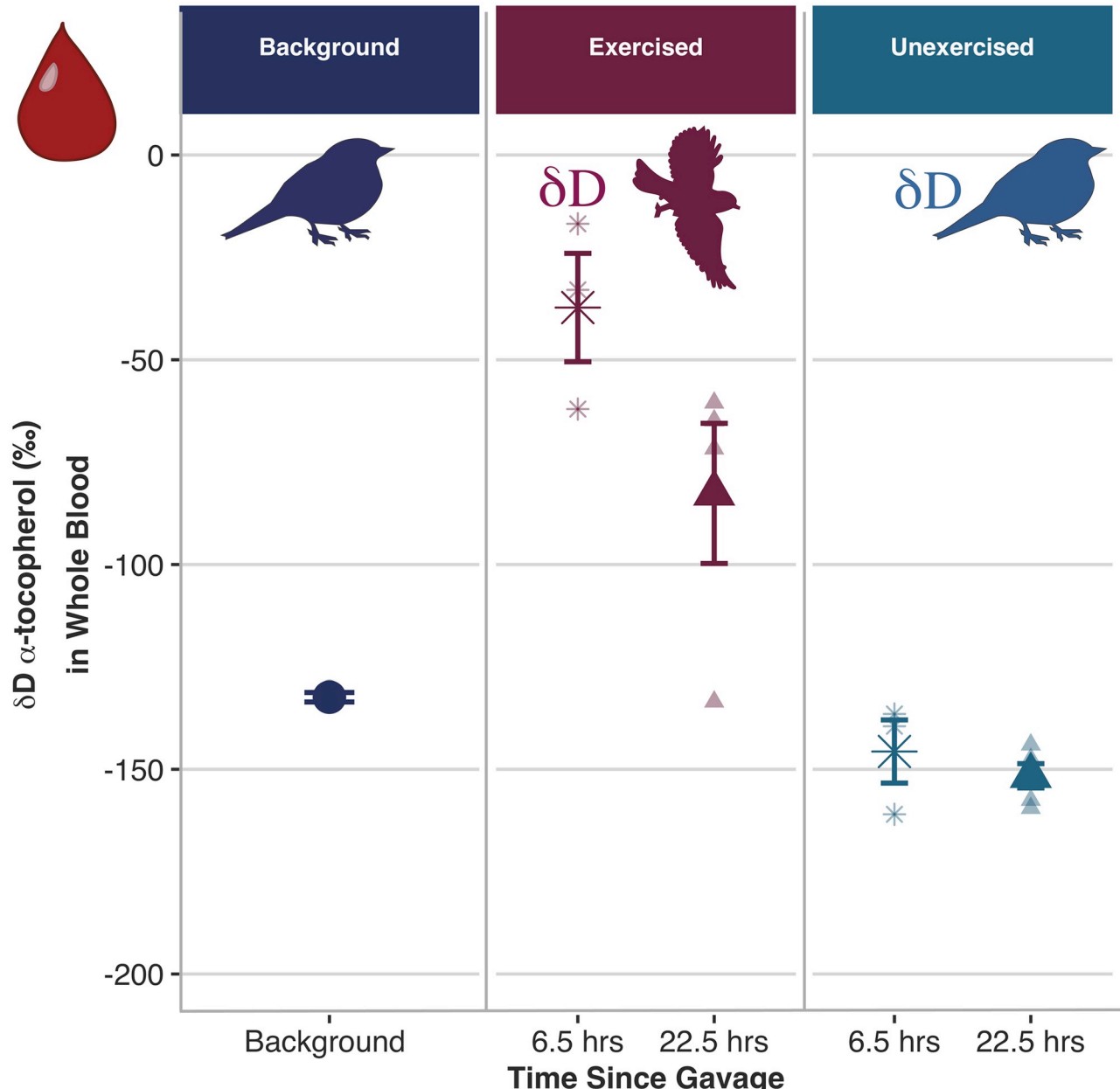

**Fig 1. Whole blood δD-α-tocopherol values from background (dark blue circle), exercised (red) and unexercised (light blue) zebra finches.** Only exercise-trained birds had deuterium values (δD α-tocopherol) that were higher than background, and exercise-trained birds gavaged 6.5 hours prior to sampling (red star) had higher deuterium values than birds gavaged 22.5 hours prior to sampling (red triangle). Background samples were obtained from birds that were gavaged with olive oil only. Exercised and unexercised samples were obtained from birds gavaged with $d_6\alpha$-tocopherol (indicated by δD). The stable isotope ratios are reported in delta-notation as parts per thousand (‰) deviations from the international standard for deuterium (δD). δD α-tocopherol values in the blood of unexercised birds were not different from background at either sampling time point (gavaged 6.5 hours prior to sampling = light blue star; gavaged 22.5 hours prior to sampling = light blue triangle; adjusted $R^2$ = 0.79, Exercised: ß = 93.45, $t_{3,14}$ = 5.03, $P < 0.001$; Unexercised: ß = 8.59, $t_{3,14}$ = 0.46, $P = 0.65$; Hours Since Gavage: ß = -1.55, $t_{3,14}$ = -2.18, $P = 0.05$). Lighter colored points represent the raw data, points and error bars represent mean ± SE.

blood converged on a value of 24.72 ± 12.76 by 180+ hrs after gavage and the median residence time, or half-life (ln(2) *τ), of gavaged δD α-tocopherol in whole blood of exercise-trained zebra finches was estimated at 17.1 hrs. In other words, 90% of gavaged δD α-tocopherol that appeared in the blood would be replaced or deposited into tissues within 2.4 days.

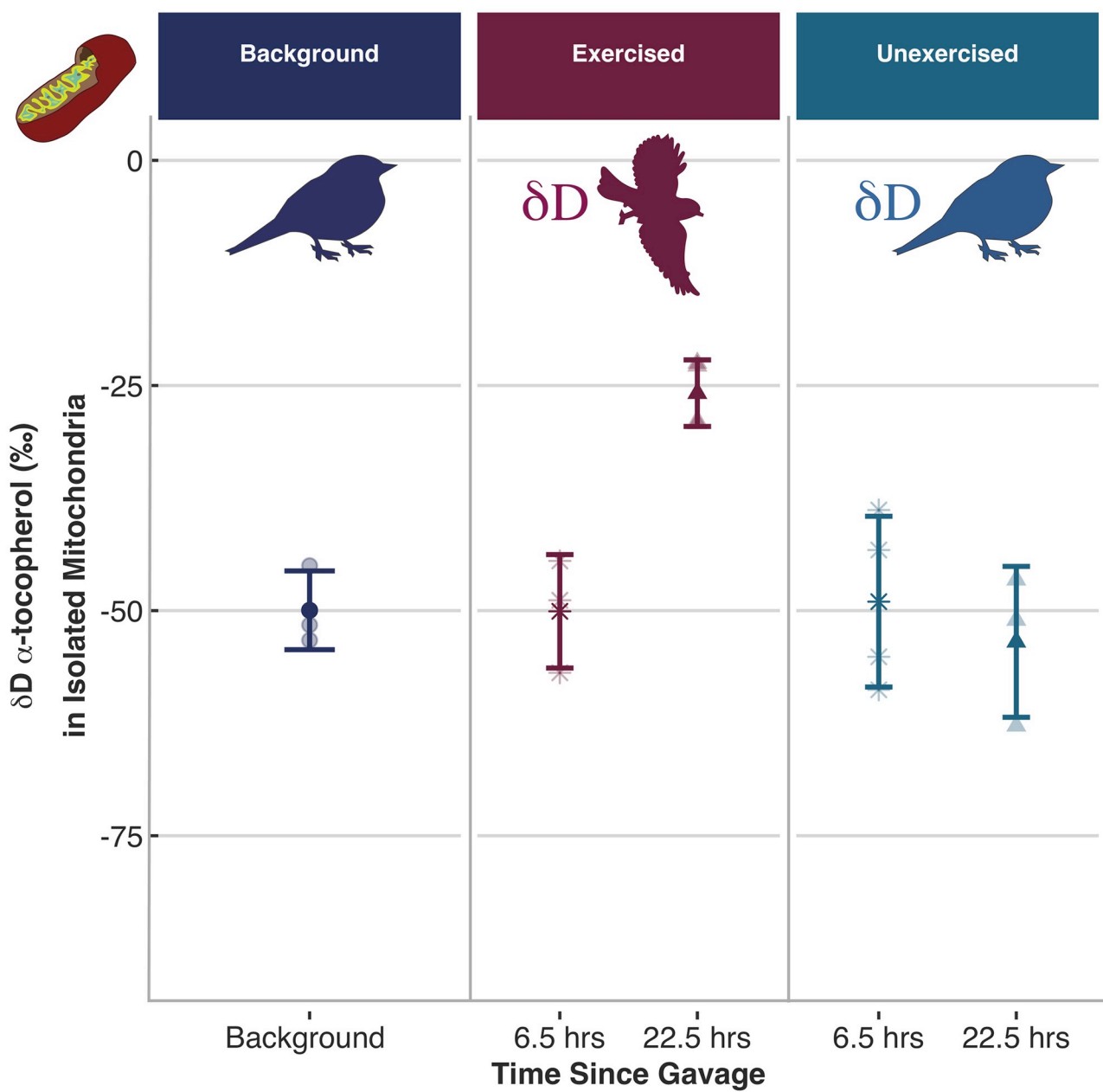

**Fig 2. δD α-tocopherol values of mitochondria isolated from the pectoral muscles of background (dark blue circle), exercised (red) and unexercised (light blue) zebra finches.** Only exercise-trained birds had deuterium values that were higher than background, and exercise-trained birds gavaged 6.5 hours prior to sampling (red star) had lower deuterium values than birds gavaged 22.5 hours prior to sampling (red triangle). Background samples were obtained from birds that were gavaged with olive oil only. Exercised and unexercised samples were obtained from birds gavaged with $d_6\alpha$-tocopherol (indicated by δD). The stable isotope ratios are reported in delta-notation as parts per thousand (‰) deviations from the international standard for deuterium (δD). δD α-tocopherol values in the blood of unexercised birds were not different from background at either sampling timepoint (gavaged 6.5 hours prior to sampling = light blue star; gavaged 22.5 hours prior to sampling = light blue triangle; adjusted $R^2 = 0.70$, Interaction: ß = 1.80, $t_{3,10} = 3.65$, $P = 0.004$; Exercised: ß = -12.73, $t_{3,10} = -1.60$, $P = 0.15$; Hours Since Gavage: ß = -0.28, $t_{3,10} = -0.80$, $P = 0.44$). Lighter colored points represent the raw data, points and error bars represent mean ± SE.

## Discussion

We show for the first time that an ingested lipophilic antioxidant reached the mitochondria in the flight muscles of a songbird and that uptake, transport and deposition of this dietary

antioxidant depended on exercise. Specifically, δD-α-tocopherol was found in the blood and mitochondria isolated from the pectoralis muscle of zebra finches within 6.5 and 22.5 hrs, respectively, but only if the birds were exercise-trained. These results indicate that exercise facilitated the absorption and deposition of α-tocopherol likely in response to increased RS production associated with intense exercise [3], and/or by inducing muscle cells to actively produce more α-tocopherol rich mitochondria [77–79]. As the most important physiological function of α-tocopherol is as an antioxidant against free radical-mediated lipid peroxidation [80, 81], the idea that increased RS production associated with exercise facilitated the absorption and deposition of α-tocopherol seems likely. However, since birds have complex endogenous antioxidant systems [3] and may consume a whole host of dietary antioxidants that can interact and impact the function of a given antioxidant [82], future studies are required to determine the exact physiological role of α-tocopherol deposited into the blood and muscle mitochondria. The absence of δD-α-tocopherol levels above baseline in unexercised birds suggests that any baseline RS generation or that associated with short flights within the cages could be neutralized via endogenous antioxidants (enzymes, sacrificial molecules, or stores of dietary antioxidants), or were in low enough of a dose to act solely as cellular messengers rather than cause oxidative stress [2].

Although δD-α-tocopherol was detected in whole blood from exercise-trained zebra finches within 6.5 hours after gavage, deposition of α-tocopherol into pectoral muscle mitochondria took more than 6.5 hrs and less than 22.5 hrs. We estimated that 50% of ingested α-tocopherol in whole blood would be replaced within ~17 hrs, and 90% replaced within 2.4 days in exercised finches. This residence time of α-tocopherol in the blood of exercised zebra finches is shorter than in the plasma of humans given deuterium-labeled foods (half-life = ~ 30 hrs), and faster than uptake into human erythrocytes [83]. Many more such studies of residence time of α-tocopherol across a broader suite of species and body sizes are required before any conclusions can be made about the allometry of such metabolism [60]. However, if our estimates of residence time for zebra finches apply to similar-sized migratory passerines, then the implication is that α-tocopherol, found in preferred fruits such as Arrowwood Viburnum (*Viburnum dentatum*) at fall migration stopover sites [13], would be available in the muscle mitochondria of migratory birds within 22.5 hours after ingestion. In other words, α-tocopherol in fruits eaten by birds today would be available the next night during (nocturnal) migration to defend against RS production in their muscle mitochondria. Although we did not directly verify that α-tocopherol, once transported to the blood and muscle mitochondria, protected against lipid peroxidation, many other studies in fowl and humans provide evidence that the main action of α-tocopherol *in vivo* is as an antioxidant [39, 84]. Future studies on migratory birds are needed that examine the metabolism of dietary antioxidants and the efficacy of α-tocopherol to protect against oxidative damage especially during exercise.

What remains to be determined is the exact mechanism by which exercise-training in small songbirds enables α-tocopherol to be absorbed into the blood and then transported to the mitochondria, although our exercise-training regime seems a reasonable approach. A number of physiological changes are associated with exercise including blood flow, energy metabolism, associated RS production, and transport of substrates to fuel exercise, all of which are reasonable processes for further investigation. We find particularly intriguing two non-mutually exclusive hypotheses related to such a mechanism: (1) exercise upregulates features of the gut (e.g., digestive enzymes, nutrient transporters, microbial communities) that enhance absorption of α-tocopherol to blood; (2) exercise increases fat metabolism which then facilitates metabolism and transport of dietary antioxidant into muscle mitochondria. Birds rely primarily on fatty acid oxidation to fuel the demands of intense exercise and, in the wild, mechanisms associated with fatty acid transport (e.g. fatty acid translocase/CD36 or plasma

membrane-bound fatty acid binding protein) and fatty acid oxidation (e.g. activation of PPARs, enzymes such as β-hydroxyacyl-Coenzyme-A dehydrogenase or carnitine palmitoyl transferase) are upregulated prior to seasonal migration [85–87]. Further, migratory flights are fueled by high levels of circulating triglycerides transported by very low density lipoproteins (VLDLs) [21]. Since α-tocopherol is lipid soluble and is transported into cells and organelles via VLDLs [57], increased transport of fatty acids and fatty-acyl CoA into muscle mitochondria for β-oxidation likely facilitates the absorption and transport of α-tocopherol into the mitochondria. Therefore, the ability to benefit from ingested antioxidants where the risk of damage by peroxyl radicals is highest (the mitochondria), could be an inevitable benefit of burning fats as fuel.

## Supporting information

**S1 File. R code for deuterium analysis.** The R code associated with the models and figures in this manuscript.
(R)

**S2 File. R code for turnover analysis.** The R code associated with the analysis of vitamin E turnover in the blood of zebra finches.
(R)

## Acknowledgments

We would first like to thank Barbara Pierce for raising the zebra finches used in this experiment, and Dr. John Lodge for providing the deuterated α–tocopherol. We are also grateful for Megan Skrip (and family) who constructed the original flight arena, and for providing guidance for animal care and flight-training throughout the experiment. Thank you to Luke "The Handler" Douglas for his cheerful help while caring for the zebra finches or conducting exercise-training. Thank you to Megan Gray and Lara Kazo for their animal care help and Chris Lane for use of his centrifuge. A big thank you to Alex Gerson and Kristen DeMoranville for valuable discussions about lipid transport and the antioxidant system during exercise in birds.

## Author Contributions

**Conceptualization:** Clara Cooper-Mullin, Scott R. McWilliams.

**Data curation:** Clara Cooper-Mullin.

**Formal analysis:** Clara Cooper-Mullin.

**Funding acquisition:** Scott R. McWilliams.

**Investigation:** Clara Cooper-Mullin, Wales A. Carter, Ronald S. Amato, David Podlesak, Scott R. McWilliams.

**Methodology:** Clara Cooper-Mullin, Scott R. McWilliams.

**Project administration:** Scott R. McWilliams.

**Resources:** David Podlesak, Scott R. McWilliams.

**Supervision:** Scott R. McWilliams.

**Validation:** Clara Cooper-Mullin.

**Visualization:** Clara Cooper-Mullin, Scott R. McWilliams.

**Writing – original draft:** Clara Cooper-Mullin.

**Writing – review & editing:** Wales A. Carter, Ronald S. Amato, David Podlesak, Scott R. McWilliams.

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
