## [Decision Letter · Decision Letter 0]

30 Nov 2020

PONE-D-20-29149

Dietary antioxidants reach the mitochondria in the flight muscle of birds but only if they exercise

PLOS ONE

Dear Dr. Cooper-Mullin,

First of all, I would like to apologize for the delay in reviewing your manuscript; it has been very challenging to find reviewers.

After careful consideration, we feel that it has merit but does not fully meet PLOS ONE’s publication criteria as it currently stands. Therefore, we invite you to submit a revised version of the manuscript that addresses the points raised during the review process.

The topic and question addressed in the manuscript are interesting. The two reviewers differ considerably in their final recommendation, however they both provide detailed and meaningful comments and suggestions for improving the manuscript.

Rev. 1 is particularly concerned about the methodology used in bird training. Specifically, if only the trained birds were handled and startled by a handler, there could be potential problems due to a conspicuous difference in stress exposure between the trained and non-trained birds, which could affect the results. Also, important concerns were raised about the statistical analyses. I agree that some analyses should be rethought, and more details need to be provided, especially about the models and variables used. Please make sure to take into account all the comments of the reviewers (including those highlighted on the PDF attached by Rev. 2) before resubmitting your paper. 

We look forward to receiving your revised manuscript.

Kind regards,

Nicoletta Righini, PhD

Academic Editor

PLOS ONE

Journal Requirements:

3. Please include a copy of Table 1 which you refer to in your text on page 17.

**Additional Editor Comments (if provided):**

Other details:

1) Please make sure to use **LINE NUMBERS** (and page numbers) in the next version of the manuscript

2) First paragraph of the Introduction: ‘Supplementing the diets of athletes (humans, birds, or mammals) with certain antioxidants….’. Sounds odd (bird and mammal athletes?). I would phrase it differently: ‘Supplementing the diets of humans (e.g., athletes), birds, or mammals with certain antioxidants….’

**Reviewers' comments:**

Reviewer's Responses to Questions

**Comments to the Author**

1. Is the manuscript technically sound, and do the data support the conclusions?

Reviewer #1: Partly

Reviewer #2: Yes

2. Has the statistical analysis been performed appropriately and rigorously? 

Reviewer #1: No

Reviewer #2: Yes

3. Have the authors made all data underlying the findings in their manuscript fully available?

Reviewer #1: Yes

Reviewer #2: Yes

4. Is the manuscript presented in an intelligible fashion and written in standard English?

Reviewer #1: Yes

Reviewer #2: Yes

5. Review Comments to the Author

**Reviewer #1:** PONE-D-20-29149

This study shows that zebra finches fed with vitamin E have higher levels of this vitamin both in their blood and muscle mitochondria when doing regular flights, but not when not exercising. Although the question is good, I have several important concerns related to study design, sampling and statistical analyses. I hope that my comments help the authors to improve their MS.

The short title is better, more explicit than the full title. Please change the full title to something similar with the short title, i.e. stating clearly that it is all about vitamin E in particular, not dietary antioxidants in general. Also, please change in the title “of birds” to “of zebra finches” because we do not know yet whether the results found in this study apply generally to all birds with very different physiological trait, adaptations related to flight and life histories.

Vitamin E reaches mitochondria in pectoral muscle cells only if zebra finches use their flight muscles. However, oxidative stress is also generated by flight muscle only if the birds fly. Therefore, the main message of the study is not very surprising.

The authors are right that a vitamin with antioxidant properties should first reach its site of activity. The second filter is to serve indeed antioxidant function once reaching the site of activity. Would have been worth to test this second assumption, i.e. whether the vitamin E that reaches mitochondria in flight muscles do reduce the amount of oxidative damage to lipids in these cells. This could have been tested by a 2-by-2 treatment design in which half of the birds in both exercise groups (trained vs. untrained) would have access to dietary vitamin E, while the other half would have received a diet deficient in vitamin E, and then measuring the amount of PUFA damage in flight muscle from birds of all 4 groups (trained-vitE, trained-deficient, untrained-vitE and untrained-deficient). Please add a paragraph in the Discussion, which discusses this limitation of the study and highlight possible avenues for future studies that can fill this gap.

Studies in exercising mammals (including humans) showed that orally administered vitamin E has very low absorption efficiency in contrast with vitamin E injected directly into the muscle. What is known about the absorption efficiency of dietary vitamin E in birds? Also, studies in mammals found that acute increase in vitamin E has beneficial effects at high altitude, but might have adverse effect at sea level. Some birds, as the zebra finch occupy low altitude habitats and are sedentary never reaching high altitudes during their flight, while other birds are migratory with much higher flight muscle activity (and potentially higher exercise-related oxidative demand) and they frequently do their journey at several thousand meters elevation. What is known about these factors in birds? This could be shortly discussed in the Discussion because it is all about the relevance or implications of the results of this study.

I missed the line numbers in the MS; without line numbers it is harder to anchor the concerns and issues raised by the reviewers.

- Introduction, first paragraph: not all reactive species are deleterious, since at low levels they serve very important and vital signalling functions, which should be recognised (i.e. RS are not just evil)

- Introduction, “protects against oxidative damage to lipids and muscles”: lipids are macromolecules, while muscle is a tissue – please clarify if damage to muscle refers to muscle proteins or muscle cell membrane lipids. This is relevant because vitamin E protects against peroxidative damage to lipids, but not against the oxidative damage of proteins.

- Introduction, “birds reliance on fats as fuel”: birds also use carbohydrates to fuel flight, especially the energetically highly costly burst flights, which partly explain why birds have very high blood glucose levels. In addition, it should be explained in more detail here why the preponderance of fats as fuel are an important factor that demand high uptake of dietary antioxidants. Typo: “birds’ reliance” instead of “birds reliance” (apostrophe missing)

- Introduction, “oxidatively vulnerable fuel”: does the authors assume that birds ingest high amount of food rich in antioxidants to protect the lipids in their food? Isn’t protecting the flight muscle cells the primary adaptive value as the scope of the study implies? Please clarify this or remove the vulnerability of dietary fats to oxidative damage.

- Introduction, “but evidence that these antioxidants protect against damage by RS”: protect what and where? Please be more explicit and clear.

- Introduction, “Importantly, if and only if dietary antioxidants are absorbed and transported to the mitochondria, can they effectively alleviate the oxidative costs associated with energy-demanding life events such as flying.”: in this sentence the authors claim that dietary non-enzymatic antioxidants are the only way to protect against flight-induced RS, which is in contradiction with the previous sentence admitting that the antioxidant system is complex. Please review here the evidence on the role of dietary non-enzymatic antioxidants AND the non-dietary gene-encoded enzymatic antioxidants in alleviating flight-induced oxidative stress.

- Introduction and throughout the MS: please use consistently either vitamin C or ascorbic acid

- Introduction, “elucidating the pace and extent of metabolic routing of a commonly consumed dietary antioxidant would reveal the efficacy of such dietary sources for paying these oxidative costs”: this claim is not in match with the results of this study and hence should be removed or toned down at least – the results show that dietary vitamin E is channelled towards muscle mitochondria but this study do not show results on whether this accumulation indeed reduces oxidative damage to muscle cell lipids. In addition, this claim is also too far-fetching because vitamin E works in tandem with vitamin C, but the authors did not measure the vitamin C content of muscle mitochondria. Does vitamin C also accumulate in mitochondrial inner membrane?

- Introduction, hypotheses: these hypotheses apparently came across a posteriori and hence match the results. They are also redundant with the study questions formulated beforehand. Please either formulate these hypotheses in light of evidence in the primary literature or remove them entirely.

- Introduction, cited references: 70 (!) references are cited already in the Introduction. This is an exaggeration. Please follow the best practice in citing literature and avoid inflating the reference list. For instance, the very first sentence of the MS is backed with seven references (refs 1–7). Citing ref 1 is absolutely sufficient here because this sentence is very general, now a biological common sense, and has noting related to exercise-induced oxidative stress (ref 2 does not add anything to ref 1, ref 3 is irrelevant here, refs 4–7 are related to exercise, but the sentence is not about exercise). Please do this pruning of unnecessary citations throughout the MS.

- Methods, second sentence: too much information compressed in this sentence – please remove the irrelevant information and provide more explicit detail on the relevant information that are retained, e.g. age distribution of birds involved in the study, their diet prior to the study, etc., details that have implications for the study

- Methods, “Importantly, zebra finches are passerines with similar genetic antioxidant regulatory pathways to wild migratory songbirds [80].”: why is this important, given that the study is about the absorption of dietary antioxidants and does not deal with genetic antioxidants? I guess the authors would like to suggest that the zebra finch is a good model organism for migratory birds and hence results on zebra finches can be generalized to migratory birds in general. However, zebra finches are not migratory and therefore might possess different physiological and muscular adaptations than the migratory species. If so, the results of this study might not necessarily apply to all migratory birds. This limitation should be admitted in the Discussion stating clearly that further studies on migratory birds are necessary to better understand the metabolism of dietary antioxidants in relation to flight demands.

- Methods, “The finches (n = 65)…”: please provide details on the age distribution of the two treatment groups, as well as of males and females. Also, please specify what the diet regime of birds was prior to the exercise treatment, i.e. while being involved in the lipid turnover rates.

- Methods, “Each day during the 48 or 70 days…”: not clear, please clarify – were the birds exercising for 48 or 70 days?

- Methods, exercise training: As I understand, exercise trained birds were removed from the aviaries and moved into the flight arena. Am I right? Were the non-trained birds handled in the same way as the trained ones and for the same amount of time with the exception that they were not allowed to burst fly? Handling stress can influence the absorption of antioxidants and the levels of other antioxidants with which vitamin E works in tandem, thus adding bias to the results. Also, exercise birds were startled by a handler for two hours per day for 48 days. Does a handler walked around the cages of non-trained birds as well for two hours per day for 48 days? Please provide these details on the handling of all 65 birds prior to exercise flights. If handling and startling stress was missing in non-trained birds, then the two groups differed not only in treatment but also in stress exposure. If so, the two effects (exercise treatment and stress exposure) cannot be separated and the results can be biased. Please note that there are several exercise training studies in which exercising birds do the exercise voluntarily without being startled by a handler and hence without being exposed to a chronic stress stimulus, see e.g. Costantini et al. 2012 Behav Ecol Sociobiol 66: 1195–1199. In my view, this is a serious shortcoming of the study unless the authors does not have strong arguments against this concern.

- Methods, “Blood sampling…” subheading, first sentence: why unbalanced sampling and not 8 birds from both treatment groups? 4 birds is a quite low sample size. Why females were not considered? A few lines below, 3 males were selected after 70 days of training. Again, very small sample size. Why untrained birds were not selected after 70 days? Why females were not considered at this time point again?

- Methods, “Blood sampling…” subheading: Why the authors did not collect pre-treatment blood samples? Blood sampling does not need the birds to be sacrificed and it would allow testing the effect of exercise treatment on changes in physiological parameters both within and among treatment groups. This is way more informative about the effect of treatment than having only post-treatment samples. In case the treatment groups had a minor difference in the measured parameters already at the pre-treatment time point (despite the random allocation of birds into treatment groups), this can either lead to significant group differences, which otherwise would not be there (type I error, false positive) or lead to non-significant differences, which otherwise would be significant (type II error, false negative), depending on in which direction the groups differ at the pre-treatment time point. This weakness of the sampling protocol makes the results related to blood samples precarious. I don’t know what the practice is with pectoral muscle sampling, so just asking: surgical biopsy from a small subset of birds is not feasible in order to have pre-treatment measurements on mitochondria as well? At the “Sample preparation” subheading the authors state that only 150 μg of muscle sample yields sufficient signal. Based on my shallow experience in small songbird surgery, this can be collected by birds being anaesthetised with ketamine-xylazine.

- Methods, Sample preparation, “Blood samples (n = 18)…”: n = 16 instead? n = 18 does not match the information a few lines above “8 trained and paired untrained birds”. Please clarify.

- Methods, Sample preparation, “… two muscle samples…”: why only two samples out of 16? Credible inference from two samples is impossible. What the remaining 14 muscle samples was used for? Were these used for isotope ratio mass spectrometry? I can’t figure out the sample sizes in the Methods. I recommend the authors prepare a small and informative table on sample collection and analyses, so the reader can unambiguously find out how many samples were collected, and how these samples were allocated for different lab analyses.

- Methods, Statistics, “Akaike’s Information Criterion for small sample sizes (Δ AICc < 2)”: this is incorrect. AICc is the abbreviation for Akaike’s Information Criterion for small sample sizes, while Δ AICc < 2 is related to model selection based on AICc, showing that two alternative models that differ in modulus AICc by less the 2 are not statistically different, i.e. their fit to the data are similarly good. Please reformulate. Also, Akaike weights are not explained in sufficient detail and some readers might not understand what these weight are meant for. Most importantly, I can’t see the importance of Information Theory-based model selection and that of Akaike weights given the very simple structure of the full model (three predictors). I would suggest a Likelihood Ratio-based model comparison of alternative models that differ in the presence/absence of one fixed effect.

- Methods, Statistics, “We report the results of the best model for each tissue (Table 1).”: please report the statistics for each model. As far as I understand, there were only three explanatory variables (treatment, time, dose), so there are only 8 alternative models (one only intercept model, three single predictor models, three two-predictor models and one three-predictor model). Showing the stats for 8 alternative models does not require much space. Or this should be provided as a supplementary table at the least. Importantly, Table 1 is not provided in the MS and hence cannot be assessed by the reviewers.

- Methods, Statistics: much more details are needed here. What was the distribution of dependent variables? Were the variances of the dependent variables homogenous for the two treatment groups? How many levels the explanatory variables have, are they categorical or continuous (OK, treatment is clearly a factor with two levels, but what about the other predictors)? If some predictors have multiple levels or are continuous, were non-linear (e.g. quadratic) effects assessed? Both females and males were used in the study. There are frequently sex-differences in physiological attributes. Please add sex as two-levels factor to the models and also asses its second-order interaction with the other predictor variables. Please add the second-order interaction of treatment as well. For instance, treatment might have different affects according to time elapsed since gavaged or alpha-tocopherol dose. Unfortunately, these models with many parameters might face convergence problems due to the small sample sizes in this study.

Please specify the housing of birds in aviaries. In how many aviaries were the birds housed? Importantly, birds housed in the same aviary are not independent samples. Therefore, aviary ID should be added as random effect to the models and thus a mixed-effects model should be used. What proportion of variance in the dependent variable is explained by the random factor, i.e. aviary effect?

Were the samples analysed in the lab by personnel who were blind to the treatment, i.e. did not know whether the samples were from treated (exercise) or control (untrained) birds. How was the repeatability (intraclass correlation coefficient) of parameters measured in duplicate or triplicate, and what was their coefficient of variation (CV%)? What R packages were used for different statistics? Please cite all the used packages.

- Results: The authors mention Information Theory approach in the Statistics section, while in the Results section we see P-values, i.e. a frequentist approach. How these statistics were found, i.e. what R function was used? Given that the model estimates are betas, I guess a type III sum of squares approach was used for significance testing. Does the order of the predictors in the model affect the results of the best model?

- Results, first statistics in the first parentheses, after Fig. 1: it is not clear what the results show here. The statistics for trained and untrained birds show that trained birds have different alpha-tocopherol levels are the two time points, while the untrained ones have similar levels at the two time points? I thought based on the Title and Introduction of the study that the main question is related to the comparison of trained vs. untrained birds. Therefore, here the statistics for this comparison is need for both time points. Also, these statistics suggest that interactions were also tested, while the Statistics section of the Methods does not tell anything about interactions.

- Discussion, “… the first explanation seems likely…”: the authors should tone their statements such to match the results of the study. For instance, this conclusion is highly speculative because the authors did not measure lipid peroxidation either from blood or muscle tissue. Elevated levels of antioxidants does not tell anything about the oxidative stress state of the birds, please see e.g. Monaghan et al. 2009 Ecol Lett 12: 75–92. So, please tone much down these and similar statements and clearly admit the limitations of the study here in the Discussion.

- Discussion, “It also indicates that birds, like mammals…”: what indicates this? This conclusion is very far fetching; please delete the entire sentence.

- Discussion, “However, the implication for migratory passerines is that vitamin E, available in preferred fruits such as Arrowwood Viburnum (Viburnum dentatum) at fall migration stopover sites [22], would be available in their muscle mitochondria within 22.5 hours after ingestion.”: is there any study in literature showing that vitamin E from natural sources (e.g. fruits) do absorb similarly to when purified alpha-tocopherol is orally administered? If not, please remove this apparently speculative sentence. Additionally, results found in captive, inbreed and non-migratory zebra finches are not necessarily applicable to free-living migratory songbirds, which clearly have different physiological adaptations due to their migratory lifestyle.

- Discussion, “In other words, vitamin E in fruits eaten by birds today would be available the next night during (nocturnal) migration to defend against reactive species production in their muscle mitochondria.”: redundant because every reader knows what less than 22.5 hours mean, and again overstated because this study does not show anything about protection against reactive oxygen species in muscles.

- References: Extremely long reference list. Most papers are cited once in the MS. This list should be considerably reduced by retaining only the most important and relevant references after each statement. Based on the length of the MS, the Reference list can be reduced to 50 references at most.

- potential relevant references: Guglielmo et al. 2011 J Exp Biol 204: 2683–2690; Costantini 2008 Ecol Lett 11: 1238–1251; Costantini et al. 2013 J Exp Biol 216: 2213–2220; Costantini et al. 2012 Behav Ecol Sociobiol 66: 1195–1199; Skrip et al. 2015 Ecol Evol 5: 3198–3209; Levin et al. 2017 Science 355: 733–735

**Reviewer #2:** 

The experiments described in the manuscript were novel, interesting and conducted with robust design and analysis. The largest problem that requires addressing in this manuscript is the writing itself. There are multiple instances of informal, hyperbolic language that is inappropriate for a peer-reviewed publication. There are several points that require additional clarification throughout the manuscript that are necessary for a non-expert reader to understand the study. These points are highlighted/commented on in the attached copy of the manuscript. One general point that should be addressed is that because tocopherol uptake mechanisms in the gut and muscle are poorly understood, it is possible that differences in uptake may be due to differences in blood flow between rest and exercise, rather than training. This may be a limitation in experimental design that is worth addressing in the manuscript.

6. PLOS authors have the option to publish the peer review history of their article (what does this mean?). If published, this will include your full peer review and any attached files.

Reviewer #1: No

Reviewer #2: No

---

## [Author Response · Author response to Decision Letter 0]

31 Mar 2021

Rebuttal Letter for PONE-D-20-29149: Dietary vitamin E reaches the mitochondria in the flight muscle of zebra finches but only if they exercise

Reviewer #1: 

This study shows that zebra finches fed with vitamin E have higher levels of this vitamin both in their blood and muscle mitochondria when doing regular flights, but not when not exercising. Although the question is good, I have several important concerns related to study design, sampling and statistical analyses. I hope that my comments help the authors to improve their MS.

Thank you for the favorable and insightful reviews of our article. We have carefully thought through your comments and responded below. We have carefully considered all of your concerns and revised the original manuscript as we addressed these concerns. Below in bold are our responses to each comment and suggestion for revision. The line numbers associated with specific changes to our manuscript refer to the revised unmarked version of our paper without tracked changes. 

The short title is better, more explicit than the full title. Please change the full title to something similar with the short title, i.e. stating clearly that it is all about vitamin E in particular, not dietary antioxidants in general. Also, please change in the title “of birds” to “of zebra finches” because we do not know yet whether the results found in this study apply generally to all birds with very different physiological trait, adaptations related to flight and life histories.

We thank the reviewer for his comment and have changed the full title accordingly.

Vitamin E reaches mitochondria in pectoral muscle cells only if zebra finches use their flight muscles. However, oxidative stress is also generated by flight muscle only if the birds fly. Therefore, the main message of the study is not very surprising.

Although we too find the story relatively straightforward, and some readers may anticipate the result, ours is the first study to document that the absorption and deposition of dietary antioxidants in birds and how it depends on exercise. Additionally, the literature on whether birds experience ‘oxidative stress’ during flight is not conclusive, especially since we are only able to measure the outcome of reactive species generation (changes to antioxidant capacity and byproducts of oxidative damage). Therefore, prior to this study, it was unclear whether a bird regardless of exercise could consume vitamin E and deposit that vitamin E into muscle mitochondria. 

The authors are right that a vitamin with antioxidant properties should first reach its site of activity. The second filter is to serve indeed antioxidant function once reaching the site of activity. Would have been worth to test this second assumption, i.e. whether the vitamin E that reaches mitochondria in flight muscles do reduce the amount of oxidative damage to lipids in these cells. This could have been tested by a 2-by-2 treatment design in which half of the birds in both exercise groups (trained vs. untrained) would have access to dietary vitamin E, while the other half would have received a diet deficient in vitamin E, and then measuring the amount of PUFA damage in flight muscle from birds of all 4 groups (trained-vitE, trained-deficient, untrained-vitE and untrained-deficient). Please add a paragraph in the Discussion, which discusses this limitation of the study and highlight possible avenues for future studies that can fill this gap.

Although we agree that examining the lipid peroxidation levels in the isolated mitochondria would be a relevant future step, there are several reasons why we did not tackle this issue in this experiment. 1) Previous work on vitamin E in mammals and birds have highlighted that it primarily acts as an antioxidant in vivo and therefore we were more interested in the bioavailability of this dietary antioxidant rather than its antioxidant action per se. 2) There are also inherent difficulties when exposing mitochondria during the isolation process to oxygen in ambient air as that itself can induce lipid peroxidation. Thus, we felt the necessary first step was to demonstrate that Vitamin E reaches the site of activity (in the mitochondria. In response to the reviewer’s comment, we have added this idea to the discussion (Lines 314-320):

“As the most important physiological function of α-tocopherol is as an antioxidant against free radical-mediated lipid peroxidation [79,80], the idea that increased RS production associated with exercise facilitated the absorption and deposition of �-tocopherol seems likely. However, since birds have complex endogenous antioxidant systems [3] and may consume a whole host of dietary antioxidants that can interact and impact the function of a given antioxidant [81], future studies are required to determine the exact physiological role of α-tocopherol deposited into the blood and muscle mitochondria.”

Studies in exercising mammals (including humans) showed that orally administered vitamin E has very low absorption efficiency in contrast with vitamin E injected directly into the muscle. What is known about the absorption efficiency of dietary vitamin E in birds? 

Absorption efficiency of vitamin E is unknown in passerine birds. Therefore, it was a major reason we chose to do this study with labeled vitamin E so that whatever was absorbed could be traced to the blood and mitochondria. We specifically chose to gavage our birds because we wanted to have the process of obtaining vitamin E from food be as similar as experimentally possible to the experience a bird would have of consuming it in the wild, digesting and processing it, before transporting it to organs, tissues, and organelles. 

Also, studies in mammals found that acute increase in vitamin E has beneficial effects at high altitude, but might have adverse effect at sea level. Some birds, as the zebra finch occupy low altitude habitats and are sedentary never reaching high altitudes during their flight, while other birds are migratory with much higher flight muscle activity (and potentially higher exercise-related oxidative demand) and they frequently do their journey at several thousand meters elevation. What is known about these factors in birds? This could be shortly discussed in the Discussion because it is all about the relevance or implications of the results of this study.

Although the effectiveness of vitamin E as an antioxidant at different altitudes is an interesting and relevant avenue of study, here we were more interested in the effect of flight on the absorption and deposition of vitamin E. Therefore, the altitude question is outside of the scope of this study. 

I missed the line numbers in the MS; without line numbers it is harder to anchor the concerns and issues raised by the reviewers.

Somehow line numbers were not included in the original submission and so we have added them to this revised version of the manuscript. Apologies for this omission.

- Introduction, first paragraph: not all reactive species are deleterious, since at low levels they serve very important and vital signalling functions, which should be recognised (i.e. RS are not just evil)

We thank the reviewer for this comment, and never meant to imply that RS are all bad. We have added a sentence to clarify this (Line 45):

“When produced at low levels, RS act as important signaling molecules in a variety of physiological processes, but also interact with mitochondria, cells and tissues to cause oxidative damage [1,2].”

- Introduction, “protects against oxidative damage to lipids and muscles”: lipids are macromolecules, while muscle is a tissue – please clarify if damage to muscle refers to muscle proteins or muscle cell membrane lipids. This is relevant because vitamin E protects against peroxidative damage to lipids, but not against the oxidative damage of proteins.

The studies that we are referring to here measured a variety of indices of oxidative damage in plasma. Some measured lipid damage (e.g. MDA), while Watson et al 2005 measured creatine kinase levels that indicate muscle damage. To avoid this confusion about damage to lipids and muscles, we split up the reference list within the sentence to clarify our meaning (Lines 52-54):

“Supplementing the diets of athletes (humans, birds, or mammals) with certain antioxidants prior to high-intensity exercise training protects against oxidative damage to lipids [8,9] and muscles [10].” 

- Introduction, “birds reliance on fats as fuel”: birds also use carbohydrates to fuel flight, especially the energetically highly costly burst flights, which partly explain why birds have very high blood glucose levels. 

We thank the reviewer for this comment and agree that birds may use a mixture of fuels during flights although not primarily carbohydrates. Experiments that examined the cost of perch to perch (or stop and go) flight in Zebra Finches have found respiratory quotients of 0.79 ± 0.01 (Hambly et al., 2002) for flights of ~2 minutes, and 0.75 ± 0.01 for longer periods of perch-to-perch flights (Nudds and Bryant, 2001, 2002); an RQ = 1.0 is indicative of carbohydrates. These RQ results indicate that Zebra Finches exposed to flight training similar to that used in this study (an hour of training at a time) used a mixture of fuel types (lipid and protein), but primarily are burning lipids (added to Lines 146-149):

“This type of short-burst flight incurs energetic costs that are approximately 3x higher than sustained flight [21] and respiratory quotients for zebra finches exposed to perch-to-perch flights were indicative of primarily burning fats (0.75 ± 0.01) [47], a fuel type that increases the potential for oxidative damage [28].”

In addition, it should be explained in more detail here why the preponderance of fats as fuel are an important factor that demand high uptake of dietary antioxidants. 

We thank the reviewer for this comment and have added more detail to the introduction (Lines 66-70):

“Fats, especially polyunsaturated fats, are particularly vulnerable to oxidative damage due to their low oxidative potential and once damaged form lipid radicals that can then attack other lipids, creating a cascade of damage [28]. Therefore, storing and catabolizing fats for flight puts birds at a higher risk of oxidative damage and an increased need for antioxidants (endogenous and/or dietary).”

Typo: “birds’ reliance” instead of “birds reliance” (apostrophe missing)

Thank you. We have fixed this in the manuscript. 

- Introduction, “oxidatively vulnerable fuel”: does the authors assume that birds ingest high amount of food rich in antioxidants to protect the lipids in their food? Isn’t protecting the flight muscle cells the primary adaptive value as the scope of the study implies? Please clarify this or remove the vulnerability of dietary fats to oxidative damage.

We thank the reviewer for this comment. This section is to point out all the potential risks of consuming fats as well as burning them during flights. Therefore, yes, birds may consume antioxidants alongside fats to protect those fats during digestion, they may store antioxidants with fats stores to protect those fats, and they may be used in other areas of high RS production (such as the mitochondria) to protect the structural fats in the different tissues and/or organelles. Therefore, we have left this section as is. 

- Introduction, “but evidence that these antioxidants protect against damage by RS”: protect what and where? Please be more explicit and clear.

Since it is unknown exactly what and where dietary antioxidants can protect against RS in wild birds, we cannot provide further clarification. In fact, addressing this unknown was part of our motivation to conduct this study in the manner that we did.

- Introduction, “Importantly, if and only if dietary antioxidants are absorbed and transported to the mitochondria, can they effectively alleviate the oxidative costs associated with energy-demanding life events such as flying.”: in this sentence the authors claim that dietary non-enzymatic antioxidants are the only way to protect against flight-induced RS, which is in contradiction with the previous sentence admitting that the antioxidant system is complex. Please review here the evidence on the role of dietary non-enzymatic antioxidants AND the non-dietary gene-encoded enzymatic antioxidants in alleviating flight-induced oxidative stress.

Since this study focuses on the absorption and deposition of dietary antioxidants, we are purposefully avoiding too much focus on bird’s endogenous sources of antioxidants. We carefully reviewed this sentence and we do not think it indicates that dietary antioxidants are the sole source of antioxidants. Rather, this sentence points out that unless dietary antioxidants are indeed absorbed and transported to the mitochondria, they cannot be effective at protecting against damage. In response to the reviewer’s comment, we have added a clarifying sentence (Line 79): 

“Further, however common in their diets, the bioavailability of dietary antioxidants in songbirds is largely unknown (although see: [37]).”

Additionally, we address the existence and ability of endogenous antioxidants to respond to damage earlier in the introduction (Line 47):

“Although animals have a complex endogenous antioxidant system that can and does respond to oxidative challenges [3], upregulating this endogenous system likely costs energy and time [4].”

- Introduction and throughout the MS: please use consistently either vitamin C or ascorbic acid

 We thank the reviewer for this comment and have changed all uses to vitamin C for consistency. 

- Introduction, “elucidating the pace and extent of metabolic routing of a commonly consumed dietary antioxidant would reveal the efficacy of such dietary sources for paying these oxidative costs”: this claim is not in match with the results of this study and hence should be removed or toned down at least – the results show that dietary vitamin E is channelled towards muscle mitochondria but this study do not show results on whether this accumulation indeed reduces oxidative damage to muscle cell lipids. In addition, this claim is also too far-fetching because vitamin E works in tandem with vitamin C, but the authors did not measure the vitamin C content of muscle mitochondria. Does vitamin C also accumulate in mitochondrial inner membrane?

We agree with the reviewer that vitamin E works in tandem with vitamin C, although the location of where vitamin C accumulates in tissues or organelles in passerines is largely unknown (although in humans there is evidence that it is transported to the inner mitochondrial membrane as DHA via GLUT-1; see Mandl et al. 2009). However, the role of vitamin C in the context of this manuscript is only to recycle the vitamin E after it interacts with a radical. Since the antioxidant properties of vitamin E have been described as its major role in vivo, we respectfully disagree with the reviewer and maintain that showing the absorption and deposition of vitamin E into the mitochondria means that it would likely be used to combat the oxidative costs in a given organelle or tissue. In response to the reviewer, we have replaced the term ‘efficacy’ with ‘potential’ as a way to ‘tone down’ this statement (Line 110).

Mandl J, Szarka A, Bánhegyi G (2009) Vitamin C: Update on physiology and pharmacology. Br J Pharmacol 157: 1097–1110. doi:10.1111/j.1476-5381.2009.00282.x

- Introduction, hypotheses: these hypotheses apparently came across a posteriori and hence match the results. They are also redundant with the study questions formulated beforehand. Please either formulate these hypotheses in light of evidence in the primary literature or remove them entirely.

As suggested, we have simplified the original two hypotheses to a single hypothesis derived from previous studies (Line 111): 

“Specifically, we tested the following hypothesis: The rate at which dietary antioxidants are routed to tissues depends on exercise with faster routing in flown vs. sedentary birds.”

- Introduction, cited references: 70 (!) references are cited already in the Introduction. This is an exaggeration. Please follow the best practice in citing literature and avoid inflating the reference list. For instance, the very first sentence of the MS is backed with seven references (refs 1–7). Citing ref 1 is absolutely sufficient here because this sentence is very general, now a biological common sense, and has noting related to exercise-induced oxidative stress (ref 2 does not add anything to ref 1, ref 3 is irrelevant here, refs 4–7 are related to exercise, but the sentence is not about exercise). Please do this pruning of unnecessary citations throughout the MS.

As suggested, we have pruned the references to the extent possible while still citing the most relevant literature. 

- Methods, second sentence: too much information compressed in this sentence – please remove the irrelevant inforsmation and provide more explicit detail on the relevant information that are retained, e.g. age distribution of birds involved in the study, their diet prior to the study, etc., details that have implications for the study

We thank the reviewer for this suggestion and have edited this section. The relevant information (age, diet) are provided in later sections of the methods. 

- Methods, “Importantly, zebra finches are passerines with similar genetic antioxidant regulatory pathways to wild migratory songbirds [80].”: why is this important, given that the study is about the absorption of dietary antioxidants and does not deal with genetic antioxidants? I guess the authors would like to suggest that the zebra finch is a good model organism for migratory birds and hence results on zebra finches can be generalized to migratory birds in general. However, zebra finches are not migratory and therefore might possess different physiological and muscular adaptations than the migratory species. If so, the results of this study might not necessarily apply to all migratory birds. This limitation should be admitted in the Discussion stating clearly that further studies on migratory birds are necessary to better understand the metabolism of dietary antioxidants in relation to flight demands.

We thank the reviewer for this suggestion and, as documented above, have edited this section including deletion of this quoted sentence. We thank the reviewer for holding us accountable on this point and have added a disclaimer (in italics) to this part of the discussion, as suggested (Line 334):

“However, if our estimates of residence time for zebra finches apply to similar-sized migratory passerines, then the implication is that α-tocopherol , found in preferred fruits such as Arrowwood Viburnum (Viburnum dentatum) at fall migration stopover sites [13], would be available in the muscle mitochondria of migratory birds within 22.5 hours after ingestion.”

We also added the following to the end of this same paragraph of the discussion, as suggested by the reviewer (Lines 340-345):

“Although we did not directly verify that α-tocopherol, once transported to the blood and muscle mitochondria, protected against lipid peroxidation, many other studies in fowl and humans provide evidence that the main action of α-tocopherol in vivo is as an antioxidant [38,83]. Future studies on migratory birds are needed that examine the metabolism of dietary antioxidants and the efficacy of α-tocopherol to protect against oxidative damage especially during exercise.”

- Methods, “The finches (n = 65)…”: please provide details on the age distribution of the two treatment groups, as well as of males and females. Also, please specify what the diet regime of birds was prior to the exercise treatment, i.e. while being involved in the lipid turnover rates.

All birds were the same age (hatch-year; added to Line 127). The numbers of males and females in the larger experiment are reported on Line 128. We described the diets fed to birds during the 8 weeks prior to the exercise treatment on lines 134-137.

“During those 8 weeks, birds acclimated to the aviaries, and to a standard mixed-seed diet (Abbaseed #3700, Hillside, NJ USA) composed mainly of canarygrass (Phalaris canariensis) and had ad libitum access to water, grit, and cuttlebone. All zebra finches were given fresh kale, a source of vitamin C and α-tocopherol, once weekly [49,50,58].”

- Methods, “Each day during the 48 or 70 days…”: not clear, please clarify – were the birds exercising for 48 or 70 days?

Given that extracting mitochondria requires a system that keeps the tissues alive, we sacrificed subsets of males in each treatment group at two different time points (48 and 70 days after the start of flight-training) so that extraction was consistently done as fast as possible after birds were sacrificed. The number of days a bird was exercised did not result in any differences within a treatment, so the duration of flight-training was not included in the final statistical models. We have added our reasoning to Line 159:

“Due to time constraints associated with extracting mitochondria while keeping tissues alive (see below), we sacrificed subsets of males in each treatment group at two different time points as follows.”

In addition, we added the following sentence to the end of this same paragraph (Lines 186-189):

“We detected no difference in the amount of deuterated α – tocopherol in samples from males sacrificed after 48 or 70 days of flight-training so the duration of flight-training was not included in the final statistical models.”

- Methods, exercise training: As I understand, exercise trained birds were removed from the aviaries and moved into the flight arena. Am I right? Were the non-trained birds handled in the same way as the trained ones and for the same amount of time with the exception that they were not allowed to burst fly? Handling stress can influence the absorption of antioxidants and the levels of other antioxidants with which vitamin E works in tandem, thus adding bias to the results. Also, exercise birds were startled by a handler for two hours per day for 48 days. Does a handler walked around the cages of non-trained birds as well for two hours per day for 48 days? Please provide these details on the handling of all 65 birds prior to exercise flights. If handling and startling stress was missing in non-trained birds, then the two groups differed not only in treatment but also in stress exposure. If so, the two effects (exercise treatment and stress exposure) cannot be separated and the results can be biased. Please note that there are several exercise training studies in which exercising birds do the exercise voluntarily without being startled by a handler and hence without being exposed to a chronic stress stimulus, see e.g. Costantini et al. 2012 Behav Ecol Sociobiol 66: 1195–1199. In my view, this is a serious shortcoming of the study unless the authors does not have strong arguments against this concern.

Thank you for the opportunity to provide more details about the experimental methods. We have added information to this section of the methods that makes clear how the exercise training was administered, and exactly the differences in handling of the flight-trained and untrained birds. We also cited five previous studies, including some of our own, that used the same set-up and that provide detailed explanations of the flight arena and how the flight-training proceeded. In short, the untrained birds were caught and administered the deuterated vitamin E at the same time as the trained birds on a given experimental day. However, we did not want to expose the untrained birds to a stimulus that would cause them to fly more (such as having an individual walk around their cages as well) and opted instead for a system that allowed them to move as wanted within their aviaries while we were in the flight arena with the flight-trained birds. Additionally, we had no indication that birds were stressed in our experiment, and there is little evidence that handling stress (or in this case having a handler walk behind the birds to encourage them to fly) alters measurements of the antioxidant system or oxidative damage (see: Costantini et al., 2007), and therefore shouldn’t necessarily change how a dietary antioxidant would be allocated due to exercise. 

- Methods, “Blood sampling…” subheading, first sentence: why unbalanced sampling and not 8 birds from both treatment groups? 4 birds is a quite low sample size. Why females were not considered? A few lines below, 3 males were selected after 70 days of training. Again, very small sample size. Why untrained birds were not selected after 70 days? Why females were not considered at this time point again?

We thank the reviewer for this comment. Although the larger group of zebra finches that we pulled from for this study did indeed include both sexes, we wanted to avoid possible differences in sexes (as found in an earlier study by Skrip et al. 2016 - cited in main manuscript) so we only used males (as explained in the first paragraph of the ‘Blood sampling” section of the Methods: Lines 156-158). In order to be able to extract the mitochondria as soon as possible after sacrificing the birds and keep the process consistent and effective, we were only able to sample up to 12 birds on a given day. This required that we sampled some males after 48 days and others 70 days into flight-training, as described above. Please note that the total number of males sampled per group was indeed balanced with 3 background birds, 8 trained birds, and 8 control individuals (total n = 19), as reported on Lines 161-165 and 167-170. 

- Methods, “Blood sampling…” subheading: Why the authors did not collect pre-treatment blood samples? Blood sampling does not need the birds to be sacrificed and it would allow testing the effect of exercise treatment on changes in physiological parameters both within and among treatment groups. This is way more informative about the effect of treatment than having only post-treatment samples. In case the treatment groups had a minor difference in the measured parameters already at the pre-treatment time point (despite the random allocation of birds into treatment groups), this can either lead to significant group differences, which otherwise would not be there (type I error, false positive) or lead to non-significant differences, which otherwise would be significant (type II error, false negative), depending on in which direction the groups differ at the pre-treatment time point. This weakness of the sampling protocol makes the results related to blood samples precarious. I don’t know what the practice is with pectoral muscle sampling, so just asking: surgical biopsy from a small subset of birds is not feasible in order to have pre-treatment measurements on mitochondria as well? At the “Sample preparation” subheading the authors state that only 150 μg of muscle sample yields sufficient signal. Based on my shallow experience in small songbird surgery, this can be collected by birds being anaesthetised with ketamine-xylazine.

We thank the reviewer for this comment. We have used the suggested before/after sampling in previous studies in which we used blood sampling to assess the effect of flight-training on the antioxidant system of zebra finches (e.g., Skrip et al. 2016, Cooper-Mullin et al. 2019 - both cited in the main manuscript). However, the present study examines the effect of flight exercise on the deposition of vitamin E into muscle mitochondria with the blood measures providing evidence of initial absorption and delivery to circulation. For small birds like Zebra finches it is not possible to take an adequate sample of muscle (0.4 g is needed to obtain 150 µg of mitochondria) via biopsy prior to flight and not potentially compromise their ability to fly. We were also concerned that anesthesia like ketamine-xylazine, required prior to biopsy, can alter the oxidative balance of an individual, which could affect the absorption and deposition of dietary antioxidants. For these reasons, we adopted a safer, more prudent sampling scheme that involved comparisons of treatment groups rather than trying to take repeated measures from the same individual. Importantly, the amount of variation in baseline deuterium values in blood or muscle is low (blood: -132.41 ± 1.16, mitochondria: -49.96 ± 2.52) indicating that background deuterium values are consistent across individuals (Reported on Line 222). 

- Methods, Sample preparation, “Blood samples (n = 18)…”: n = 16 instead? n = 18 does not match the information a few lines above “8 trained and paired untrained birds”. Please clarify.

We thank the reviewer for this comment, and in fact, this was a typo as it should say n = 19. Originally, we took blood and muscle samples from 3 background birds, 8 trained birds, and 8 control individuals (total n = 19) as described above and is now clarified in the revised manuscript. Some samples were unable to be analyzed and we report final sample sizes at line 220:

“Final sample sizes varied after analysis as some samples were not able to be dried completely (Blood Samples: n = 3 Background, 7 Trained, 8 Untrained; Mitochondria Samples: n = 3 Background, 7 Trained, 7 Untrained).”

- Methods, Sample preparation, “… two muscle samples…”: why only two samples out of 16? Credible inference from two samples is impossible. What the remaining 14 muscle samples was used for? Were these used for isotope ratio mass spectrometry? I can’t figure out the sample sizes in the Methods. I recommend the authors prepare a small and informative table on sample collection and analyses, so the reader can unambiguously find out how many samples were collected, and how these samples were allocated for different lab analyses.

We apologize for the confusion here, as the two muscle samples were used only as a preliminary analysis to determine how much mitochondria we would need to successfully extract in order to see a deuterium signal (a common, but often unreported aspect of these types of experiments). Given this study was part of a larger experiment focused on lipid turnover rates and changes in the antioxidant system in exercised and non-exercised birds (Carter et al., 2018 and Cooper-Mullin et al. 2019), we were limited in the amount of muscle we were allocated for this study. Therefore, we prioritized extracting mitochondria from the muscle samples in order to examine whether the vitamin E was deposited into the mitochondria, not just the larger muscle tissue infrastructure. To avoid confusion, we have reworded this section (Line 196): 

“We used 150 µg of blood and mitochondria based on an earlier analysis that showed sufficient δD α-tocopherol appeared in muscle tissue with this amount of sample.”

 Additionally, we report the final samples sizes for all the blood samples after all lab analyses at line 220:

“Final sample sizes varied after analysis as some samples were not able to be dried completely (Blood Samples: n = 3 Background, 7 Trained, 8 Untrained; Mitochondria Samples: n = 3 Background, 7 Trained, 7 Untrained).”

- Methods, Statistics, “Akaike’s Information Criterion for small sample sizes (Δ AICc < 2)”: this is incorrect. AICc is the abbreviation for Akaike’s Information Criterion for small sample sizes, while Δ AICc < 2 is related to model selection based on AICc, showing that two alternative models that differ in modulus AICc by less the 2 are not statistically different, i.e. their fit to the data are similarly good. Please reformulate. Also, Akaike weights are not explained in sufficient detail and some readers might not understand what these weight are meant for. Most importantly, I can’t see the importance of Information Theory-based model selection and that of Akaike weights given the very simple structure of the full model (three predictors). I would suggest a Likelihood Ratio-based model comparison of alternative models that differ in the presence/absence of one fixed effect.

We thank the reviewer for the opportunity to clarify our model selection procedures - the revised text from the “Statistics” section of the methods is copied below (Lines 233-241). Although there are always many different approaches to analyzing this type of experimental data, the simple structure of the full model (flight-trained vs. untrained treatment groups) and the need to control for potential effects of ‘time since gavage’ and small variation in ‘amount of gavage’ made a model comparison approach like that suggested by the reviewer not necessary. 

“We used the information-theoretic approach based on Akaike’s Information Criterion for small sample sizes (AICc) to select the best model for each tissue (blood, muscle). The highest ranked models were those with the highest Akaike weight (wi) and a change in AICc (ΔAICc) [73]. Akaike weights represent the weight of evidence in favor of a given model being the best model and given that one of the set of candidate models must be the “best” model based on the distance (Kullback–Leibler information) between the selected model and the next candidate model [73]. We obtained coefficients (ß), t-values, and p-values for each individual coefficient for each top model with the function summary in the base R stats package ([74], (Table 1).”

- Methods, Statistics, “We report the results of the best model for each tissue (Table 1).”: please report the statistics for each model. As far as I understand, there were only three explanatory variables (treatment, time, dose), so there are only 8 alternative models (one only intercept model, three single predictor models, three two-predictor models and one three-predictor model). Showing the stats for 8 alternative models does not require much space. Or this should be provided as a supplementary table at the least. Importantly, Table 1 is not provided in the MS and hence cannot be assessed by the reviewers.

We apologize that Table 1 was missing from our original submission. It is now included in the manuscript and it shows the results of the top two candidate models as well as the intercept-only model for comparison. 

- Methods, Statistics: much more details are needed here. What was the distribution of dependent variables? Were the variances of the dependent variables homogenous for the two treatment groups? How many levels the explanatory variables have, are they categorical or continuous (OK, treatment is clearly a factor with two levels, but what about the other predictors)? 

Thank you for the opportunity to provide more details about our statistical methods. We have added more details to the statistical methods (as outlined above), and more details on homogeneity of variance, specifically, to Lines 241-246: 

“We also assessed whether non-linear effects would improve the models, but found that they did not provide a better model fit. We checked whether variables met the assumptions of homogeneity of variance with Levene’s tests using the car package (v_3.0-9, [75]), visually inspected residual plots, and did not find any noticeable deviations from homoscedasticity or normality. Results are presented as mean ± standard error (SE) and the full R code is available in supplementary materials.”

You are correct that treatment was treated as a factor, time after gavage was treated as continuous, as was the amount of administered δD α-tocopherol. We have added this information to the methods section. 

If some predictors have multiple levels or are continuous, were non-linear (e.g. quadratic) effects assessed? 

We thank the reviewer for making sure our statistics are robust. We assessed whether non-linear effects would improve the models (as outlined above), but found that they did not provide a better model fit and therefore kept the original linear regressions.

Both females and males were used in the study. There are frequently sex-differences in physiological attributes. Please add sex as two-levels factor to the models and also asses its second-order interaction with the other predictor variables. 

We thank the reviewer for this comment. Although the larger group of zebra finches that we pulled from for this study did indeed include both sexes, we wanted to avoid possible differences in sex so we only used males (as explained in the section on blood sampling, mitochondrial isolation and mass spectrometry and clarified at line 156):

“We randomly drew 19 males from the larger flight-trained and untrained groups of zebra finches (n = 65) for this experiment. We sampled only males because a previous study found differences in oxidative damage associated with flight-training between males and females [28].”

Please add the second-order interaction of treatment as well. For instance, treatment might have different affects according to time elapsed since gavaged or alpha-tocopherol dose. 

Unfortunately, these models with many parameters might face convergence problems due to the small sample sizes in this study.

We hope that clarifying our statistical methods and results through the above comments have answered these questions. 

Please specify the housing of birds in aviaries. In how many aviaries were the birds housed? Importantly, birds housed in the same aviary are not independent samples. Therefore, aviary ID should be added as random effect to the models and thus a mixed-effects model should be used. What proportion of variance in the dependent variable is explained by the random factor, i.e. aviary effect?

We thank the reviewer for this comment. Although the larger population of birds were housed in 4 aviaries, the birds used for this experiment were housed in only 2 aviaries (one for males in the exercise-trained treatment, one for males in the untrained treatment). Therefore, the aviaries they were housed in were the same as the treatment and including aviary number in the model is unnecessary. Additionally, it is important to note that although they were housed in separate aviaries, they were all in the same room, and birds were able to see across aviaries as they were made of wire mesh. We have clarified this in the manuscript (Line 131): 

“The aviaries were made of wire mesh and birds were provided perches for natural movement.” 

Were the samples analysed in the lab by personnel who were blind to the treatment, i.e. did not know whether the samples were from treated (exercise) or control (untrained) birds. How was the repeatability (intraclass correlation coefficient) of parameters measured in duplicate or triplicate, and what was their coefficient of variation (CV%)? 

Although mitochondrial isolation was performed by the same individual with knowledge of the metadata associated with each sample, on a given day samples were all processed at once. The lab in Los Alamos, however, was blind to the treatment until after all samples were run on the mass spectrometer. All CVs were less than 10% (added to line 219):

“All samples were run in triplicate and coefficients of variation for each sample were less than 10%.” 

What R packages were used for different statistics? Please cite all the used packages.

We thank the reviewer for this comment and have now cited all R packages in our methods section. 

- Results: The authors mention Information Theory approach in the Statistics section, while in the Results section we see P-values, i.e. a frequentist approach. How these statistics were found, i.e. what R function was used? Given that the model estimates are betas, I guess a type III sum of squares approach was used for significance testing. Does the order of the predictors in the model affect the results of the best model?

We obtained coefficients (ß), t-values, and p-values for each individual coefficient for each top model with the summary function in the base R stats package. This way the order of predictors in the model did not affect the results of the best model. We have added this information to the methods section (Line: 239) 

- Results, first statistics in the first parentheses, after Fig. 1: it is not clear what the results show here. The statistics for trained and untrained birds show that trained birds have different alpha-tocopherol levels are the two time points, while the untrained ones have similar levels at the two time points? I thought based on the Title and Introduction of the study that the main question is related to the comparison of trained vs. untrained birds. Therefore, here the statistics for this comparison is need for both time points. Also, these statistics suggest that interactions were also tested, while the Statistics section of the Methods does not tell anything about interactions.

We hope that our expanded methods that more fully explain the statistical analyses and our revised explanation of the results clear up any confusion. However, it is worth noting that since we are examining the enrichment of blood or mitochondria from the label (deuterium) on the vitamin E, it is always important to compare to the background levels of deuterium. Flight trained birds had deuterium values above that of the background birds, but untrained birds did not. Therefore, we can say that flight training effected whether or not birds absorbed vitamin E and deposited that vitamin E into their muscle mitochondria. 

- Discussion, “… the first explanation seems likely…”: the authors should tone their statements such to match the results of the study. For instance, this conclusion is highly speculative because the authors did not measure lipid peroxidation either from blood or muscle tissue. Elevated levels of antioxidants does not tell anything about the oxidative stress state of the birds, please see e.g. Monaghan et al. 2009 Ecol Lett 12: 75–92. So, please tone much down these and similar statements and clearly admit the limitations of the study here in the Discussion.

We thank the reviewer for their careful consideration of the implications of our study. We agree that the presence of vitamin E in the mitochondria does not necessarily indicate that birds are experiencing oxidative stress. However, previous work has indicated that birds actively adjust their antioxidant system and/or experience oxidative damage during flight and therefore, we do not think it is unreasonable to suggest that the potential reason vitamin E is routed to the site of RS production (muscle mitochondria) is to prevent oxidative damage during flight. However, we have added a statement to the discussion to further qualify this statement (Line 317):

“However, since birds have complex endogenous antioxidant systems [3] and may consume a whole host of dietary antioxidants that can interact and impact the function of a given antioxidant [81], future studies are required to determine the exact physiological role of α-tocopherol deposited into the blood and muscle mitochondria.” 

- Discussion, “It also indicates that birds, like mammals…”: what indicates this? This conclusion is very far fetching; please delete the entire sentence.

We thank the reviewer for this comment and have deleted this sentence from the manuscript. 

- Discussion, “However, the implication for migratory passerines is that vitamin E, available in preferred fruits such as Arrowwood Viburnum (Viburnum dentatum) at fall migration stopover sites [22], would be available in their muscle mitochondria within 22.5 hours after ingestion.”: is there any study in literature showing that vitamin E from natural sources (e.g. fruits) do absorb similarly to when purified alpha-tocopherol is orally administered? If not, please remove this apparently speculative sentence. Additionally, results found in captive, inbreed and non-migratory zebra finches are not necessarily applicable to free-living migratory songbirds, which clearly have different physiological adaptations due to their migratory lifestyle.

- Discussion, “In other words, vitamin E in fruits eaten by birds today would be available the next night during (nocturnal) migration to defend against reactive species production in their muscle mitochondria.”: redundant because every reader knows what less than 22.5 hours mean, and again overstated because this study does not show anything about protection against reactive oxygen species in muscles.

Part of our rationale for doing this experiment was to determine whether consumed dietary antioxidants (in this case Vitamin E) reached the muscle mitochondria where RS are primarily produced because such a demonstration for birds has potentially broad ecological relevance. As the reviewer has discerned, our use of zebra finches as a model system for such work has merit, but requires us to qualify our statements about the implications of the results for wild birds. The other reviewer requested more discussion of the ecological relevance of the results, so we have chosen to retain this brief section of the Discussion but appropriately qualify these statements in the revised manuscript. Here is the revised section of the Discussion (Lines 334-345): 

“However, if our estimates of residence time for zebra finches apply to similar-sized migratory passerines, then the implication is that α-tocopherol, found in preferred fruits such as Arrowwood Viburnum (Viburnum dentatum) at fall migration stopover sites [13], would be available in the muscle mitochondria of migratory birds within 22.5 hours after ingestion. In other words, α-tocopherol in fruits eaten by birds today would be available the next night during (nocturnal) migration to defend against reactive species production in their muscle mitochondria. Although we did not directly verify that α-tocopherol, once transported to the blood and muscle mitochondria, protected against lipid peroxidation, many other studies in fowl and humans provide evidence that the main action of α-tocopherol in vivo is as an antioxidant [38,83]. Future studies on migratory birds are needed that examine the metabolism of dietary antioxidants and the efficacy of α-tocopherol to protect against oxidative damage especially during exercise.”

- References: Extremely long reference list. Most papers are cited once in the MS. This list should be considerably reduced by retaining only the most important and relevant references after each statement. Based on the length of the MS, the Reference list can be reduced to 50 references at most.

As suggested, we have pruned the references to the extent possible while still citing all relevant literature.

- potential relevant references: Guglielmo et al. 2011 J Exp Biol 204: 2683–2690; Costantini 2008 Ecol Lett 11: 1238–1251; Costantini et al. 2013 J Exp Biol 216: 2213–2220; Costantini et al. 2012 Behav Ecol Sociobiol 66: 1195–1199; Skrip et al. 2015 Ecol Evol 5: 3198–3209; Levin et al. 2017 Science 355: 733–735

We thank the reviewer for these suggested references and have included them when and where appropriate. 

Reviewer #2: 

The experiments described in the manuscript were novel, interesting and conducted with robust design and analysis. The largest problem that requires addressing in this manuscript is the writing itself. There are multiple instances of informal, hyperbolic language that is inappropriate for a peer-reviewed publication. There are several points that require additional clarification throughout the manuscript that are necessary for a non-expert reader to understand the study. These points are highlighted/commented on in the attached copy of the manuscript. 

Thank you for the favorable and insightful reviews of our experiment. We have carefully considered all of your concerns and revised the original manuscript as we addressed these concerns. Below in bold are our responses to each comment and suggestion for revision. The line numbers associated with specific changes to our manuscript refer to the unmarked version of our paper without tracked changes. There are a number of suggestions made by this reviewer (including this general comment about our writing style) that has to do with how we portray the context of why we did this work. We assume that the readers of PLOS ONE are a very broad and mostly scientific audience who might enjoy reading papers with this type of (less formal) rhetoric. 

One general point that should be addressed is that because tocopherol uptake mechanisms in the gut and muscle are poorly understood, it is possible that differences in uptake may be due to differences in blood flow between rest and exercise, rather than training. This may be a limitation in experimental design that is worth addressing in the manuscript.

We agree with the reviewer that increased blood flow is a possible mechanism associated with exercise training that could contribute to the results we found. In fact, any aspect of physiology that is altered by exercise might contribute to our results. Therefore, we have changed our wording on how we talk about these changes to be clear that we are not assuming any one mechanism underlies our results (Line 348).

“A number of physiological changes are associated with exercise including blood flow, energy metabolism, associated RS production, and transport of substrates to fuel exercise, all of which are reasonable processes for further investigation.” 

Specific Comments:

 “All animals must produce energy to fuel metabolism and an inevitable byproduct of this metabolism are reactive species (RS) produced in the mitochondria that must be quenched to avoid oxidative damage to mitochondria, cells, and tissues [1–7].”

• a long and sometime inaccurate sentence - in order to "produce energy" animals would have to contravene at least one law of thermodynamics

We thank the reviewer for this comment. We have changed this sentence to read (Line 43): 

“Mitochondria are the key sites of energy metabolism in all air breathing organisms. However, an inevitable byproduct of this metabolism are reactive species (RS) produced in the mitochondria [1].”

General Introduction: pretty vague, not much background about the nature of RS. How do RS damage cellular targets? What are the functional consequences of oxidative stress?

We thank the reviewer for this comment. Given that in this study we are primarily focused on dietary antioxidants (specifically, vitamin E), we did not want to spend much time in the introduction on the nature of RS damage. We have cited several reviews that detail these intricate relationships within our introduction. For example:

1) Skrip MM, McWilliams SR. Oxidative balance in birds: An atoms-to-organisms-to-ecology primer for ornithologists. J F Ornithol. 2016;87: 1–20. doi:10.1111/jofo.12135

2) Espinosa-Diez C, Miguel V, Mennerich D, Kietzmann T, Sánchez-Pérez P, Cadenas S, et al. Antioxidant responses and cellular adjustments to oxidative stress. Redox Biol. 2015;6: 183–197. doi:10.1016/j.redox.2015.07.008

3) Cooper-Mullin C, McWilliams SR. The role of the antioxidant system during intense endurance exercise: lessons from migrating birds. J Exp Biol. 2016;219: 3684–3695. doi:10.1242/jeb.123992

 “respond to oxidative challenges” 

• Clarify this: does the antioxidant system defend against RS? Repair RS-induced damage?

We thank the reviewer for this comment and have added that the endogenous system can either prevent or repair damage and we added a citation that tackles how this complex system works (Lines 47-49): 

“Although animals have a complex endogenous antioxidant system that can and does respond to oxidative challenges [3], upregulating this endogenous system likely costs energy and time [4].”

 “costs energy” 

• how much? providing a range of estimates could inform the reader about how important ingested antioxidants may be

We wholeheartedly agree with the reviewer that understanding the exact cost of upregulating the endogenous antioxidant system would illuminate the importance of dietary antioxidants; however, these costs are unknown for many reasons, including (but not limited to) the fact that we cannot directly measure RS damage in animals and must, instead, rely on downstream markers of that damage. Additionally, RS are important signaling molecules that can, themselves, alter energy metabolism. We agree that studies that investigate the relationships between oxidative stress, antioxidant upregulation, nutrition and energy expenditure are needed, but outside the scope of this study. 

 “nature’s most energy dense, yet oxidatively vulnerable fuel” 

• hyperbole, overstatement...hydrazine is "natural" and has way more cal/g

Although we agree that technically hydrazine has more cal/g, it is not available to or consumed by free living animals. We prefer to retain this reminder to readers that fat is an energy dense but oxidatively vulnerable fuel used by many animals. In response to the other reviewer, we have also added the following few sentences that further clarify this statement (Lines 66-70):

“Fats, especially polyunsaturated fats, are particularly vulnerable to oxidative damage due to their low oxidative potential and once damaged form lipid radicals that can then attack other lipids, creating a cascade of damage [28]. Therefore, storing and catabolizing fats for flight puts birds at a higher risk of oxidative damage and an increased need for antioxidants (endogenous and/or dietary).”

 “Due to the nature of aerodynamics, perch-to-perch flights like the flight performed while foraging increase an individual’s metabolic rate up to 30 times basal metabolic rate (BMR)”

• interesting, but begs questions...how does this type of flight compare with sustained migratory flight? if it costs more why? because of all the acceleration/deceleration?

We describe such a comparison between perch-to-perch flight and sustained flights on Lines 73-75: 

“Likewise, long-duration, sustained flights by migrating birds, some of the world’s most incredible athletes, require prolonged periods operating at high metabolic rates (>10X BMR) while not eating or drinking [30–32].”

Additionally, the energy costs of the various types of flight (take off, sustained flights, and landing) are well established in the literature. Therefore, we cite: Nudds & Bryant 2000 (The energetic cost of short flights in birds, J Exp Biol.), Tatner & Bryant 1986 (Flight cost of a small passerine measured using doubly labeled water: implications for energetics studies, Auk), as well as Wikelski et al. 2003 (Costs of migration in free-flying songbirds, Nature) for further details.

 “and, therefore, likely result in a high oxidative load” 

• so high MR = high damage? or high ROS production? some support should be provided for such a statement.

The reviewer is correct that an increase in metabolic rate likely leads to an increase in RS production, which in turn leads to a higher potential for damage. Therefore, we have changed the sentence to read (Lines 70-73): 

“Additionally, due to the nature of aerodynamics, perch-to-perch flights like the flight performed while foraging increase an individual’s metabolic rate up to 30 times basal metabolic rate (BMR) [21,29] and, therefore, likely increase RS production and impose a high oxidative load [28].”

 “some of the world’s most incredible athletes” 

• More hyperbole

Although this may seem like hyperbole, the idea that birds are incredible athletes is well established in the literature. Flight is more costly (10-30x BMR) than running or swimming and yet birds are able to sustain it for hours to days at a time. Comparatively, the best of the best human athletes (Tour de France riders) sustain metabolic rates of around 5x BMR. Therefore, we have left the phrase as is in the manuscript. 

“but evidence that these antioxidants protect against damage by RS during high-intensity exercise is murky due in part to our inability to directly measure rates of RS production in vivo [2,51–53],”

• not logically conistent...you say you want to know about DAMAGE, but it's hard because you can't measure PRODUCTION...one does not equal the other

Thank you for asking us to clarify this. We have changed the sentence to read (Line 75): 

“Birds regularly consume lipophilic and hydrophilic antioxidants from fruits during fall migration [11,13,23,33], but evidence that these antioxidants protect against damage by RS during high-intensity exercise is murky due in part to the complexity of the antioxidant system [3,28,34], as well as our inability to directly measure rates of RS production in vivo that would result in damage [35,36].” 

“Importantly, if and only if dietary antioxidants are absorbed and transported to the mitochondria, can they effectively alleviate the oxidative costs associated with energy-demanding life events such as flying.”

• "absorbed" implies a passive process, but "transported" implies active; we need to know how vitamin E is assimilated from the diet and into the cells.

• 2) this is largely self-evident, therefore superfluous

Thanks for asking us to clarify this – we were in fact trying to emphasize a twostep process where first vitamin E must be absorbed and then it must be transported. Therefore we have changed the sentence to read (Lines 81-83): 

“Importantly, if and only if dietary antioxidants are absorbed and then transported to the mitochondria, can they effectively alleviate the oxidative costs associated with energy-demanding life events such as flying.”

However, we disagree that this statement is superfluous because this statement highlights one of the reasons we did the experiment. To answer the question: “is vitamin E absorbed?”, we decided to gavage the birds with labelled α-Tocopherol and measure levels of that α-Tocopherol in the blood (as opposed to other studies that inject the labeled antioxidant into circulation). To answer the second question: “is vitamin E transported to the mitochondria?” we measured levels of this antioxidant in the mitochondria from the muscle. Since any lipophilic antioxidant cannot passively move through circulation without some sort of chaperone, the presence of the labeled α-Tocopherol in the muscle mitochondria indicates it must have been transported there. 

 “α-Tocopherol” 

• awkward to start a sentence/paragraph like this

We agree and have changed this sentence to now read (Lines 84-87): 

“One of the eight stereoisomers of vitamin E (α-Tocopherol) is an essential dietary antioxidant (i.e., cannot be produced de novo by any vertebrate and so must be consumed) that acts as a chain-breaker in the propagation of lipid peroxidation [1,28,38], and once oxidized can be recycled by vitamin C in vivo [1].”

 “de novo”

• be consistent; at some points you italicize foreign phrases but here you don't

We thank the reviewer for this observation and have corrected these inconsistencies in the manuscript. 

 “) that acts as a chain-breaker in the propagation of lipid peroxidation [1,57,58], and once oxidized can be recycled by vitamin C in vivo [59].”

• chain breaker? what does this mean? keep the language formal. Clarify this: does a-tocopherol prevent lipid peroxidation? Is a-tocopherol part of the repair system for lipid peroxides?

The term “chain-breaker” is a well-established term in the literature for how vitamin E disrupts the cycle of lipid peroxidation that is initiated when RS damages a lipid molecule. The cited literature in that sentence further explains this process in detail: Halliwell & Gutteridge J. Free Radicals in Biology and Medicine. 4th ed. Oxford University Press; 2007.; Traber & Atkinson. Vitamin E, antioxidant and nothing more. Free Radic Biol Med. 2007;; Skrip & McWilliams. Oxidative balance in birds: An atoms-to-organisms-to-ecology primer for ornithologists. J F Ornithol. 2016).

“ascorbic acid” 

• be consistent; previously you called it "vitamin C"

We thank the reviewer for this comment and have changed all to vitamin C for consistency.

“is metabolically routed” 

• what does this mean? does it mean "preferentially deposited in working muscle"? if so how would one demonstrate this?

We use the term metabolically routed to indicate the full spectrum of metabolic processes (absorption, metabolism and repackaging in the liver, and transport) of vitamin E after McCue et al. 2011 and Podlesak and McWilliams 2007 - all cited in the revised manuscript. 

“with relatively higher metabolic rates and potential reactive species production”

• "relative" to what? presumably you mean to compare with something else here

We thank the reviewer for an opportunity to clarify our writing and have changed this sentence to include what is being compared (Line 101): 

“(a) that α-tocopherol from the diet is metabolically routed [46,47] to the muscle mitochondria in a volant species with relatively higher metabolic rates and potential reactive species production than mammals or fish”

“elucidating the pace and extent of metabolic routing of a commonly consumed dietary antioxidant would reveal the efficacy of such dietary sources for paying these oxidative costs”

• no idea what this is supposed to mean

We have clarified above what the term metabolic routing means and we have replaced the word “efficacy” in the original sentence with “potential” to clarify meaning. 

“Specifically, we tested the following hypotheses: (1) Lipid-soluble dietary antioxidants are routed to the mitochondria within 2-30 hours of ingestion in agreement with previous studies on the appearance and half-life of vitamin E in plasma of livestock and humans [61,69,70]”

• #1 is actually a prediction, not hypotheses

As suggested, we have deleted the original hypothesis list and now have a single hypothesis based on the general idea from primary literature that routing of dietary antioxidants would be impacted by exercise (Line 111): 

“Specifically, we tested the following hypothesis: The rate at which dietary antioxidants are routed to tissues depends on exercise with faster routing in flown vs. sedentary birds.”

“pace” 

• do you mean "rate"?

We thank the reviewer for this comment and have changed pace to rate in the manuscript. 

“The pace of this routing of dietary antioxidants depends on exercise with faster routing in flown vs. sedentary birds”

• well sure, but would this just be a consequence of blood flow? without some background on HOW vitamin is assimilated from the diet and deposited in tissue it's hard to evaluate these predictions

It is possible that any aspect of physiology that is altered by exercise could contribute to these results. We have changed our wording on how we talk about these changes to be clear that we are not assuming any one mechanism underlies our results. 

“on zebra finches”

• Scientific name?

Thank you to the reviewer for catching this oversight. We have added the scientific name to the manuscript (Line 118). 

Methods (all): In accordance with university ethics regulations?

We thank the reviewer for this question and have added to the methods (Line 184):

“All care and experimental procedures were reviewed and approved by the University of Rhode Island’s Institutional Animal Care and Use Committee under protocol AN11-12-009.”

“The exercise-trained zebra finches were subjected daily to two one-hour periods of stop-and-go perch-to-perch flights (11:00 – 12:00 and 13:30 – 14:30) in a 6 (l) x 3 (w) x 2 m tall flight arena for up to 70 days (see [54,71,79,83,84]”

• Minimum number of exercise training days? Next sentence implies minimum is 48

Thank you for allowing us to clarify this. Yes, the minimum is 48 days. Since extracting mitochondria requires a system that keeps the tissues alive, extraction needs to be done as fast as possible after birds are sacrificed. Therefore, we devised a sampling scheme where a maximum of 12 birds were sacrificed on a given day. Therefore, some birds in the experiment were sacrificed on day 48 of exercise training and some on day 70 of exercise training. We have added our reasoning to Line 159:

“Due to time constraints associated with extracting mitochondria while keeping tissues alive (see below), we sacrificed subsets of males in each treatment group at two different time points as follows.”

Missing a closing bracket

Thank you for catching this typo. We have fixed it in the manuscript. 

“. This type of short-burst flight incurs energetic costs that are approximately 3x higher than sustained flight [31].”

• good information for the Introduction

We agree with the reviewer that this information should be in the introduction. 

 “150 µL”

• what mole quantity is this? is this possible to report?

“deuterated α–tocopherol”

We have added the mole quantity of the gavage amount (0.200 μmoles) to Line 163.

• where was the label? uniformly labelled? if not would the labelled H participate in the redox reactions?

We used to d6-labeled alpha tocopherol acetate (as indicated on Line 166), which means that 6 of the hydrogen atoms on the tocopherol ring methyl groups have been replaced with deuterium atoms with an increased mass of 6. When alpha tocopherol interacts with RS, such as a lipid hydroperoxyl, only the aromatic ring is affected (El-Beltagi et al. 2013). The aromatic ring donates a hydrogen to the RS and the result is a relatively stable free radical form of d6-labeled alpha tocopherol acetate. Therefore, the labeled H do not participate in the redox reactions. 

El-Beltagi HS, Mohamed HI. Reactive oxygen species, lipid peroxidation and antioxidative defense mechanism. Not Bot Horti Agrobot Cluj-Napoca. 2013;41: 44–57. doi:10.15835/nbha4118929

“This latter group was used to determine the natural background level of deuterated hydrogen in the blood and isolated mitochondria of zebra finches and is generally assumed to be highly stable”

• Olive oil gavage was performed only in untrained. Is it assumed that natural background is similar in trained birds?

Yes, background levels of deuterium should be similar in trained and untrained birds. Therefore, we felt comfortable using only untrained birds to obtain background levels. 

took a blood sample from the severed carotid artery, and removed the right pectoral muscle 

• anaesthesia & euthanasia?

Thanks for asking us to clarify this – birds were sacrificed without anesthesia (as that could affect the viability of the mitochondria) by cervical dislocation in accordance with our IACUC protocol. We have added this information to the methods (Line 175): 

“Directly after the second one-hr flight training, we sacrificed the trained and untrained birds by cervical dislocation, took a blood sample from the severed carotid artery, and removed the right pectoral muscle.”

“at 80oC”

• Typo: missing "-"

Thank you. We have fixed this in the manuscript.

“Mitochondria” 

• this word is the plural

We thank the reviewer for catching our typo and have fixed this in the manuscript. 

“run”

• more formal language

Thank you for making sure our methods are clear. We have changed this to “analyzed with” in the manuscript. 

“Upon arrival in New Mexico, blood and mitochondria samples were also allowed to equilibrate with local air and freeze dried again for 24 hours on non-consecutive days (48 hours total).” 

• why was this necessary? the curious reader will want to know.

Standard procedure for measures of hydrogen isotopes is to allow equilibration of samples with local air because of potential exchange with ambient air that varies in background hydrogen isotope composition across the globe. We have added a citation (Vander Zandin et al. 2016) to this sentence for the curious readers (Line 193).

“delta values”

• most people don't understand these values

We thank the reviewer for making sure our manuscript is as accessible to as wide of an audience as possible and clarified how we measure delta values (Line 216):

“Stable isotope ratios are reported in delta-notation as parts per thousand (‰) deviations from the international standard for deuterium (δD). Therefore, sample delta values are a delta scale calibrated to the VSMOW standard of 2H/1H = 0.00 ± 0.3.”

“both 6.5 hrs and 22.5 hrs”

• is this time following gavage?

Yes. We have clarified this in the manuscript. 

“uptake and metabolism” 

• well, not really...you showed that the incorporation depended on exercise history...this experiment could not determine rates of uptake or "metabolism" of the vitE

We agree that this jargon may be confusing so we have changed the sentence to read (Line 307):

“We show for the first time that an ingested lipophilic antioxidant reached the mitochondria in the flight muscles of a songbird and that uptake, transport and deposition of this dietary antioxidant depended on exercise.”

“These results indicate that exercise facilitated the absorption and deposition of vitamin E” 

• this is highly accurate

“likely due to a need for external antioxidants to combat increased reactive species production associated with intense exercise” 

• this, on the other hand is wishful thinking...what does "need" have to do with it? we need to know how the uptake happens in order to understand how it was impacted by exercise

We appreciate the reviewer’s careful review of our wording and have changed this sentence to make our point clear (Line 311): 

“These results indicate that exercise facilitated the absorption and deposition of �-tocopherol likely in response to increased RS production associated with intense exercise [3], and/or by inducing muscle cells to actively produce more α-tocopherol rich mitochondria [76–78].”

Here you switched back to "reactive species" instead of "RS" - be consistent

We thank the reviewer for this comment and have changed reactive species to RS for consistency.

“As the most important physiological function of vitamin E is as an antioxidant against free radical-mediated lipid peroxidation [98,99], the first explanation seems likely.”

• which explanation? you just gave us two possibilities

We have elucidated our thought process here in the manuscript (Line 314):

“As the most important physiological function of �-tocopherol is as an antioxidant against free radical-mediated lipid peroxidation [79,80], the idea that increased RS production associated with exercise facilitated the absorption and deposition of �-tocopherol seems likely.”

If tocopherol can be 'recycled' with ascorbate, why would importing additional tocopherol be necessary?

We thank the reviewer for this interesting question. Although α-tocopherol can be recycled by vitamin C, efficient recycling requires that vitamin C be present at the exact site that tocopherol is oxidized. Since vitamin C is a hydrophilic antioxidant and is hard to measure in vivo, it is unclear whether it can be present in large quantities in the muscle mitochondria. Additionally, not all passerines can synthesize vitamin C and therefore, the ability to recycle α-tocopherol in vivo may also depend on dietary sources of vitamin C. We felt it was important to mention this potential interaction, but further discussion of how vitamin E and vitamin C interact are outside the scope of this study. 

“It also indicates that birds, like mammals, have mechanisms to maintain plasma α-tocopherol concentration [70,101], and prevent the buildup of excess α-tocopherol by preventing absorption or through efficient metabolism and excretion by the liver.”

• this is all stuff we need to know...the data from the untrained birds suggest little uptake or rapid excretion...why and how does this happen? how might exercise change this?

The other reviewer pointed out that whether birds maintain α-tocopherol in their plasma is outside the scope of this manuscript, and after careful review, we agree, so we have removed this line from the discussion. 

“Many more such studies of residence time of vitamin E across a broader suite of species and body sizes are required before any conclusions can be made about the allometry of such metabolism [83].”

• This subject is a large departure from rest of manuscript

Although we agree with the reviewer that this is not the main subject of this manuscript, we wanted to be careful to not overstate our results pertaining to how we modeled the residence time of vitamin E in the blood. 

“However, the implication for migratory passerines is that vitamin E, available in preferred fruits such as Arrowwood Viburnum (Viburnum dentatum) at fall migration stopover sites [22], would be available in their muscle mitochondria within 22.5 hours after ingestion.“ 

• Interesting point. How does 22.5h compare to stopover time in migrating passerines? Is this uptake time ecologically relevant?

Time spent on stopover during fall migration is variable depending on the species, weather, time of year, migration strategy, and on available resources for refueling. On average, however, stopovers tend to run the gamut anywhere between 1 day to several weeks (usually dependent on how quickly a bird can refuel). Therefore, 22.5 hours after ingesting berries with dietary antioxidants in them is an ecologically relevant timeframe for passerines. 

“(1) exercise upregulates features of the gut (e.g., digestive enzymes, nutrient transporters, microbial communities) that enhance absorption of α-tocopherol to blood;“ 

• great, finally we talk about this... what are the mechanisms? they must tell us what they know of if it is not known

“(2) exercise increases fat metabolism which then facilitates metabolism and transport of dietary antioxidant into muscle mitochondria.” 

• how is it transported into the cell and how could fat metabolism increase this?

“Since vitamin E is lipid soluble and is transported into cells and organelles via VLDLs” 

• Mechanistic explanations for vitamin E uptake would be beneficial in the Introduction: at the gut and at the muscle

The last three comments are all in the same vein, so we will respond as one. Since the mechanisms of uptake and transport to the cell in birds, especially passerines, is largely unknown, we did not want to confuse the reader by emphasizing these potential mechanisms in the introduction. However, we do acknowledge these knowledge gaps at Line 94: 

“Although mitochondria are known to have high α-tocopherol concentrations, especially in the inner mitochondrial membrane [44,45], the timing and deposition of α-tocopherol from the diet to plasma to mitochondria in different tissues remains largely unknown, especially in passerine birds.”

Figures: Unclear how data is summarized: are these mean/median values? What do the error bars represent? Also unclear why different symbols are used to denote different measurement timepoints. Significant differences were reported in the written component of the Results section, so they should be reported here too

Thank you for making sure our figures are clear to our readers. We have added significant differences to our figure legends and clarified that the points and error bars represent means ± standard error. We had hypotheses about how long it would take for deuterium to be absorbed and we found significant differences among time points. Therefore, we wanted to emphasize this aspect of the study by using different symbols to denote different measurement time points. 

Figure 1: need to explain y axis values/units

Figure 2: “δD-α-tocopherol values”

• what are these units?

As above, we have clarified that the units represent the amount of labeled tocopherol present in the blood or mitochondria (Line 164 and Line 216): 

“150 μL (or 0.200 μmoles) of deuterated α – tocopherol (d6-RRR- α – tocopherol acetate, a gift from Dr. John Lodge, St. Thomas’ Hospital, London, hereafter: δD α-tocopherol or d6α-tocopherol) dissolved in olive oil.”

“Stable isotope ratios are reported in delta-notation as parts per thousand (‰) deviations from the international standard for deuterium (δD).”

---

## [Decision Letter · Decision Letter 1]

10 May 2021

PONE-D-20-29149R1

Dietary vitamin E reaches the mitochondria in the flight muscle of zebra finches but only if they exercise

PLOS ONE

Dear Dr. Cooper-Mullin,

The authors have done a good job in addressing previous concerns. I especially commend them for the detailed and careful response to the queries of both reviewers. However, I concur with Reviewer 2, who still identifies a few remaining issues (see annotated PDF attached). Overall, I am pleased to recommend that the manuscript be accepted for publication pending minor revisions. 

We invite you to submit a revised version of the manuscript that addresses the points raised during the review process.

We look forward to receiving your revised manuscript.

Kind regards,

Nicoletta Righini, PhD

Academic Editor

PLOS ONE

Journal Requirements:

Reviewers' comments:

Reviewer's Responses to Questions

**Comments to the Author**

1. If the authors have adequately addressed your comments raised in a previous round of review and you feel that this manuscript is now acceptable for publication, you may indicate that here to bypass the “Comments to the Author” section, enter your conflict of interest statement in the “Confidential to Editor” section, and submit your "Accept" recommendation.

Reviewer #2: (No Response)

2. Is the manuscript technically sound, and do the data support the conclusions?

Reviewer #2: Yes

3. Has the statistical analysis been performed appropriately and rigorously? 

Reviewer #2: Yes

4. Have the authors made all data underlying the findings in their manuscript fully available?

Reviewer #2: Yes

5. Is the manuscript presented in an intelligible fashion and written in standard English?

Reviewer #2: Yes

6. Review Comments to the Author

Reviewer #2: I commend the authors for making many improvements to the manuscript since the previous round of revisions. While the manuscript is overall good, there are still some areas to address before it is suitable for publication. There are points in the main text of the manuscript that require additional clarification, explanation and consistency. In addition, the figures require additional attention, to reduce confusion and difficulty in interpretation by the reader. Please see my specific comments on the manuscript for additional details.

7. PLOS authors have the option to publish the peer review history of their article (what does this mean?). If published, this will include your full peer review and any attached files.

Reviewer #2: No

---

## [Author Response · Author response to Decision Letter 1]

24 May 2021

I commend the authors for making many improvements to the manuscript since the previous round of revisions. While the manuscript is overall good, there are still some areas to address before it is suitable for publication. There are points in the main text of the manuscript that require additional clarification, explanation and consistency. In addition, the figures require additional attention, to reduce confusion and difficulty in interpretation by the reader. Please see my specific comments on the manuscript for additional details.

Thank you for your careful review of our manuscript. We have thoroughly thought through your comments and responded below. Below in bold are our responses to each comment and suggestion for revision. The line numbers associated with your comments are associated with the previous unmarked version of the manuscript and the line numbers associated with specific changes to our manuscript refer to the current revised unmarked version of our paper without tracked changes.

Line 43: “Mitochondria are the key sites of energy metabolism in all air breathing organisms.”

• While true, this statement could be expanded to be more accurate. Fish do not breathe air yet their mitochondria are key sites of energy metabolism.

We thank the reviewer for this comment and have changed the sentence to read (Line 43): 

“Mitochondria are the key sites of energy metabolism in all oxygen consuming organisms.”

Line 58: “Although the functional importance of dietary antioxidants to animal health and performance is recognized [16–18], no previous studies have demonstrated that consumed antioxidants are transported to the primary site of reactive species production, the mitochondria, and whether this depends at all on the physiological state of the animal.”

• Some emerging evidence has indicated that mitochondria may not be as significant of a reactive species production as previously thought. Please see: Zhang, Y., & Wong, H. S. (2021). Are mitochondria the main contributor of reactive oxygen species in cells?. Journal of Experimental Biology, 224(5), jeb221606.

We thank the reviewer for pointing out this recent commentary and have changed this sentence to read (Line 56): 

“Although the functional importance of dietary antioxidants to animal health and performance is recognized [16–18], no previous studies have demonstrated that consumed antioxidants are transported to a major site of RS production, the mitochondria (but see: [19]), and whether this depends at all on the physiological state of the animal.”

Line 64: “Flight is a particularly energetically expensive form of exercise and is primarily fueled by catabolizing stored fat - nature’s most energy dense, yet oxidatively vulnerable fuel [27,28].”

• Specify here that you are referring to "metabolic" fuels.

We thank the reviewer for this comment and have clarified that we are referring to metabolic fuels (Line 64): 

“Flight is a particularly energetically expensive form of exercise and is primarily fueled by catabolizing stored fat - nature’s most energy dense, yet oxidatively vulnerable metabolic fuel [28,29].” 

Line 81: “Importantly, if and only if dietary antioxidants are absorbed and then transported to the mitochondria, can they effectively alleviate the oxidative costs associated with energy-demanding life events such as flying.”

• "assimilated by the gut" would be a preferable terminology to describe this process.

We thank the reviewer for this comment and have changed the sentence accordingly (Line 81):

“Importantly, if and only if dietary antioxidants are assimilated by the gut and then transported to the mitochondria, can they effectively alleviate the oxidative costs associated with energy-demanding life events such as flying.”

Line 103: “reactive species”

• Reactive species were defined as "RS" on line 44; stay consistent with acronyms.

We thank the reviewer for catching this oversight and have changed all mentions of reactive species after line 44 to RS. 

Lines 157-160: “We sampled only males because a previous study found differences in oxidative damage associated with flight-training between males and females [28]. Due to time constraints associated with extracting mitochondria while keeping tissues alive (see below), we sacrificed subsets of males in each treatment group at two different time points as follows.”

• This wording is unclear; is the authors' point that the sampling regime was chosen due to decline in mitochondrial quality with time?

Thank you for asking us to explain our methods. In order to properly isolate mitochondria from muscle, the tissue must be kept on ice and “alive” throughout the entire isolation procedure. Therefore, it was impossible for us to process more than 12 birds in a given day. We have changed our wording to clarify this issue (Line 160): 

“Due to time constraints associated with the need to keep the tissues cold and alive during the process of isolating mitochondria from pectoral muscle (see below), we sacrificed subsets of males in each treatment group at two different time points as follows.”

Lines 161 -165: “After 48 days of daily exercise training, we randomly selected 8 male zebra finches from the untrained group and 4 male zebra finches from the trained group to be gavaged with 150 μL (or 0.200 μmoles) of deuterated α – tocopherol (d6-RRR- α – tocopherol acetate, a gift from Dr. John Lodge, St. Thomas’ Hospital, London, hereafter: δD α-tocopherol or d6α-tocopherol) dissolved in olive oil.”

• Please be consistent here with your terminology; the next mention of the labelled tocopherol on line 165 does not follow either of these short-hand terms.

Thank you for making sure we are consistent. We have made edits where appropriate to make sure that we are consistent. Additionally, we expanded on our rationale for why we use two different terms (Lines 162-167: 

“After 48 days of daily exercise-training, we randomly selected 8 male zebra finches from the unexercised group and 4 male zebra finches from the exercised group to be gavaged with 150 μL (or 0.200 μmoles) of deuterated α – tocopherol (d6-RRR- α – tocopherol acetate, a gift from Dr. John Lodge, St. Thomas’ Hospital, London, hereafter: d6α-tocopherol or δD α-tocopherol to indicate the isotopic ratio) dissolved in olive oil.”

Line 168: “150 μL of deuterated α – tocopherol (δD α-tocopherol)”

• Unclear why this is defined again after being defined on lines 164-165.

We thank the reviewer for this comment and have changed this sentence in the manuscript (Line 169):

“After 70 days of exercise-training, we randomly selected a second group of 4 male zebra finches from the exercised group to be gavaged with 150 μL of d6α-tocopherol, and 3 male zebra finches from the unexercised group to be gavaged with 150 μL of olive oil.”

Line 177: “in vivo”

• This should be italicized

We thank the reviewer for this observation and have corrected this issue in the manuscript. 

Line 179: “took a blood sample from the severed carotid artery”

• Avian red blood cells contain mitochondria that presumably generate reactive species and likely contain antioxidants, such as a-tocopherol. How much of the a-tocopherol detected is in the VLDLs and hence available to be taken up by systemic tissues?

We entirely agree with the reviewer that understanding the amount of α-tocopherol circulating in VLDLs would be interesting. However, we did not isolate VLDLs from the whole blood to measure the amount of labeled α-tocopherol and therefore we cannot address this issue. That being said, the fact that the labeled α-tocopherol was found in the isolated mitochondria from pectoral muscle indicates that at least some of the ingested α-tocopherol was available to be taken up by tissues (at least in birds that experienced exercise).

Line 182: “Mitochondria from each bird were isolated using differential centrifugation and a Percoll gradient [67–70].” 

• Please specify here that you isolated mitochondria from the pectoralis

We thank the reviewer for this comment and have clarified this in the manuscript (Line 184): 

“Mitochondria from the pectoral muscle of each bird were isolated using differential centrifugation and a Percoll gradient [67–70].”

Line 284 (Figure 1 Legend): “Figure 1. Whole blood δD-α-tocopherol values from background (dark blue circle), exercised (red) and unexercised (light blue) Zebra Finches.”

• While these values were explained in the main text, please provide a brief explanation on the nature of these values in the figure caption. Figures and their associated captions should be interpreted independently of the main text. Please repeat for Fig. 2.

We have added an explanation of the values to the figure legends (see our response to the comment below). 

• Please be consistent with terminology; it is unclear why zebra finch is capitalized in this case and in Fig. 2, yet is not capitalized in the main text.

Thank you for keeping our language and terminology consistent. We have changed it so all mentions of zebra finches are not capitalized throughout the manuscript. 

Line 289: “Deuterium values in the blood of untrained birds were not different from background at either sampling time point”

• Similar to a previous comment on figure captions, please provide a brief explanation of what "background" means. Please repeat for Fig. 2 

Thank you for making sure our figures are clear. We have updated the figure captions to be standalone: 

“Fig 1. Whole blood δD-α-tocopherol values from background (dark blue circle), exercised (red) and unexercised (light blue) zebra finches. 

Only exercise-trained birds had deuterium values (δD α-tocopherol) that were higher than background, and exercise-trained birds gavaged 6.5 hours prior to sampling (red star) had higher deuterium values than birds gavaged 22.5 hours prior to sampling (red triangle). Background samples were obtained from birds that were gavaged with olive oil only. Exercised and unexercised samples were obtained from birds gavaged with d6α-tocopherol (indicated by δD). The stable isotope ratios are reported in delta-notation as parts per thousand (‰) deviations from the international standard for deuterium (δD). δD α-tocopherol values in the blood of unexercised birds were not different from background at either sampling time point (gavaged 6.5 hours prior to sampling = light blue star; gavaged 22.5 hours prior to sampling = light blue triangle; adjusted R2 = 0.79, Exercised: ß = 93.45, t3,14 = 5.03, P < 0.001; Unexercised: ß = 8.59, t3,14 = 0.46, P = 0.65; Hours Since Gavage: ß = -1.55, t3,14 = -2.18, P = 0.05). Lighter colored points represent the raw data, points and error bars represent mean ± SE.”

“Fig 2. δD α-tocopherol values of mitochondria isolated from the pectoral muscles of background (dark blue circle), exercised (red) and unexercised (light blue) zebra finches. 

Only exercise-trained birds had deuterium values that were higher than background, and exercise-trained birds gavaged 6.5 hours prior to sampling (red star) had lower deuterium values than birds gavaged 22.5 hours prior to sampling (red triangle). Background samples were obtained from birds that were gavaged with olive oil only. Exercised and unexercised samples were obtained from birds gavaged with d6α-tocopherol (indicated by δD). The stable isotope ratios are reported in delta-notation as parts per thousand (‰) deviations from the international standard for deuterium (δD). δD α-tocopherol values in the blood of unexercised birds were not different from background at either sampling timepoint (gavaged 6.5 hours prior to sampling = light blue star; gavaged 22.5 hours prior to sampling = light blue triangle; adjusted R2 = 0.70, Interaction: ß = 1.80, t3,10 = 3.65, P = 0.004; Exercised: ß = -12.73, t3,10 = -1.60, P = 0.15; Hours Since Gavage: ß = -0.28, t3,10 = -0.80, P = 0.44). Lighter colored points represent the raw data, points and error bars represent mean ± SE.”

Line 310: “mitochondria of zebra finches”

• Specify here that you are referring to pectoralis mitochondria.

We thank the reviewer for this comment and have clarified that we are referring to mitochondria from the pectoralis (Lines 319-321):

“Specifically, δD-α-tocopherol was found in the blood and mitochondria isolated from the pectoralis muscle of zebra finches within 6.5 and 22.5 hrs, respectively, but only if the birds were exercise-trained.”

Line 315: “the idea that increased RS production associated with exercise facilitated the absorption and deposition of α-tocopherol seems likely.”

• Please expand on this rationale: is there a mechanism that could explain how increased RS production facilitates dietary antioxidant assimilation? If so, please explain it.

Thank you for this comment. However, the focus of our paper was on the extent to which a dietary antioxidant consumed by birds was absorbed and deposited in muscle mitochondria and whether this depended on exercise. Given that our study was not designed to elucidate the mechanisms, we prefer to refrain from providing further conjecture about the mechanisms.

Line 324: “overwhelming damage”

• Somewhat hyperbolic; oxidative stress or damage would be more appropriate terminology

We thank the reviewer for this comment and have changed the sentence to read (Line 330):

“The absence of δD-α-tocopherol levels above baseline in unexercised birds suggests that any baseline RS generation or that associated with short flights within the cages could be neutralized via endogenous antioxidants (enzymes, sacrificial molecules, or stores of dietary antioxidants), or were in low enough of a dose to act solely as cellular messengers rather than cause oxidative stress [2].”

Line 330: “This residence time of α-tocopherol in the blood of trained zebra finches is more rapid than in plasma of humans given deuterium-labeled foods (half-life = ~ 30 hrs), and more rapid than uptake into human erythrocytes [82].”

• This statement requires some re-wording: 1) "residence time" is a duration, so it can be or "shorter" rather than "more rapid." 2) the term half-life here is confusing; I presume it refers to the food itself, not the deuterium (being a stable isotope) but the phrasing may confuse some readers.

Thank you for pushing us to make this manuscript as accessible as possible. We have changed the wording to “shorter” rather than “more rapid” to be consistent with a duration. Additionally, we previously define the term half life on Lines 280-283:

“Estimated retention time (τ) in whole blood converged on a value of 24.72 ± 12.76 by 180+ hrs after gavage and the median residence time, or half-life (ln(2) *τ), of gavaged δD α-tocopherol in whole blood of exercise-trained zebra finches was estimated at 17.1 hrs. In other words, 90% of gavaged δD-α-tocopherol that appeared in the blood would be replaced or deposited into tissues within 2.4 days.”

Line 356: “mechanisms associated with fatty acid oxidation”

• Do any of these mechanisms include increased fatty acid uptake? E.g. fatty acid translocase/CD36, FABP which may be more relevant to the uptake of a-tocopherol than an oxidizing enzyme like HOAD.

The reviewer is absolutely correct that fatty acid uptake is important for the uptake of tocopherol and some mechanisms associated with fatty acid uptake are also upregulated during migration. Therefore, we have added this information to the discussion (Lines 365-370): 

“Birds rely primarily on fatty acid oxidation to fuel the demands of intense exercise and, in the wild, mechanisms associated with fatty acid transport (e.g. fatty acid translocase/CD36 or plasma membrane-bound fatty acid binding protein) and fatty acid oxidation (e.g. activation of PPARs, enzymes such as β-hydroxyacyl-Coenzyme-A dehydrogenase or carnitine palmitoyl transferase) are upregulated prior to seasonal migration [85–87].”

Figures: 

• Provide some explanation as to what the "dD" symbol here and under Untrained represents. Repeat for Fig. 2

As above, we have updated the figure legends to explain all the symbols and terms.

• The y-axis title requires units. Repeat for Fig. 2

We thank the reviewer for this suggestion and have updated the figures to reflect the units as parts per thousand (‰) deviations from the international standard for deuterium. 

• Unclear why this page is landscape when all others are portrait.

This page was automatically formatted by PLOS ONE. The figures can be downloaded in high resolution for best viewing by clicking the link in the corner. 

Other: “Stable isotope ratios are reported in delta-notation as parts per thousand (‰)

deviations from the international standard for deuterium (δD).”

• This is a nice explanation that should be included in the figure captions

We are glad you think this is a good explanation and we have now included it in the figure captions.

---

## [Editor Report · Decision Letter 2]

2 Jun 2021

Dietary vitamin E reaches the mitochondria in the flight muscle of zebra finches but only if they exercise

PONE-D-20-29149R2

Dear Dr. Cooper-Mullin,

We’re pleased to inform you that your manuscript has been judged scientifically suitable for publication and will be formally accepted for publication once it meets all outstanding technical requirements.

Kind regards,

Nicoletta Righini, PhD

Academic Editor

PLOS ONE

**Additional Editor Comments (optional):**

Overall the authors have done a good job revising the manuscript, which has consideraly improved compared to the first version, thanks to the thorough revisions of the referees. I consider that the manuscript is now acceptable for publication and congratulate Dr. Cooper-Mullin and collaborators on their work.

---

## [Editor Report · Acceptance letter]

18 Jun 2021

PONE-D-20-29149R2 

Dietary vitamin E reaches the mitochondria in the flight muscle of zebra finches but only if they exercise 

Dear Dr. Cooper-Mullin:

I'm pleased to inform you that your manuscript has been deemed suitable for publication in PLOS ONE. Congratulations! Your manuscript is now with our production department. 

Kind regards, 

on behalf of

Dr. Nicoletta Righini 

Academic Editor

PLOS ONE